# IMPROVED PROBABILISTIC IMAGE-TEXT REPRESENTATIONS

**Sanghyuk Chun**
NAVER AI Lab

## ABSTRACT

Image-Text Matching (ITM) task, a fundamental vision-language (VL) task, suffers from the inherent ambiguity arising from multiplicity and imperfect annotations. Deterministic functions are not sufficiently powerful to capture ambiguity, prompting the exploration of probabilistic embeddings to tackle the challenge. However, the existing probabilistic ITM approach encounters two key shortcomings; the burden of heavy computations due to the Monte Carlo approximation, and the loss saturation issue in the face of abundant false negatives. To overcome the issues, this paper presents an improved Probabilistic Cross-Modal Embeddings (named PCME++) by introducing a new probabilistic distance with a closed-form solution. In addition, two optimization techniques are proposed to enhance PCME++ further: first, the incorporation of pseudo-positives to prevent the negative effect under massive false negatives; second, mixed sample data augmentation for probabilistic matching. Experimental results on MS-COCO Caption and two extended benchmarks, CxC and ECCV Caption, demonstrate the effectiveness of PCME++ compared to state-of-the-art ITM methods. The robustness of PCME++ is also evaluated under noisy image-text correspondences. In addition, the potential applicability of PCME++ in automatic prompt-filtering for zero-shot classification is shown. The code is available at https://github.com/naver-ai/pcmepp.

## 1 INTRODUCTION

Given images and captions, Image-Text Matching (ITM) is the task of retrieving the most relevant images/captions for the given query caption/image (Frome et al., 2013; Young et al., 2014; Kiros et al., 2014; Faghri et al., 2018; Gu et al., 2018; Lee et al., 2018; Huang et al., 2018; Li et al., 2019; Song & Soleymani, 2019; Wehrmann et al., 2019; Wu et al., 2019; Wang et al., 2020; Diao et al., 2021; Chun et al., 2021; Chen et al., 2021; Huang et al., 2021; Biten et al., 2022; Kim et al., 2023; Radford et al., 2021). The applications of ITM include cross-modal retrieval (Faghri et al., 2018) from paired image-caption datasets, such as MS-COCO Caption (Chen et al., 2015), and zero-shot classification (Radford et al., 2021), by treating class labels as a text (*e.g.*, "a photo of $\{\cdot\}$"). Owing to its significant role in image understanding and language comprehension, ITM has emerged as a fundamental Vision Language (VL) downstream task. However, this problem inherently suffers from the ambiguity caused by *many-to-many correspondences* and *sparse annotations* of the ITM datasets.

The nature of image-text matching is *many-to-many*; an image can be described in numerous text explanations, and there are a plentiful number of visual scenes to visualize a text description. However, simultaneously, our datasets are *sparsely annotated*. The existing ITM datasets are built by collecting paired image-caption and treating the collected image-caption pairs as the only positives without considering other potential positives in "negative" pairs (Chen et al., 2015; Plummer et al., 2015; Sharma et al., 2018; Changpinyo et al., 2021; Desai et al., 2021). For example, Chun et al. (2022) showed that the MS-COCO Caption dataset has massive missing positives; 88.2% of caption-to-image positives and 72.1% of image-to-caption positives are labeled as "negative", *i.e.*, false negatives, (FNs). Figure 1 shows an example. While humans judge all images and texts are plausibly matched, the dataset only treats a pair $(x_v^i, x_t^j)$ as positive when $i = j$. This paper argues that the inherent multiplicity and abudant FNs lead to the ambiguity of ITM datasets (§2.1).

This paper aims to design a proper joint embedding space that represents the inherent ambiguity by probabilistic embeddings (Oh et al., 2019), *i.e.*, encoding an input to a random variable rather than

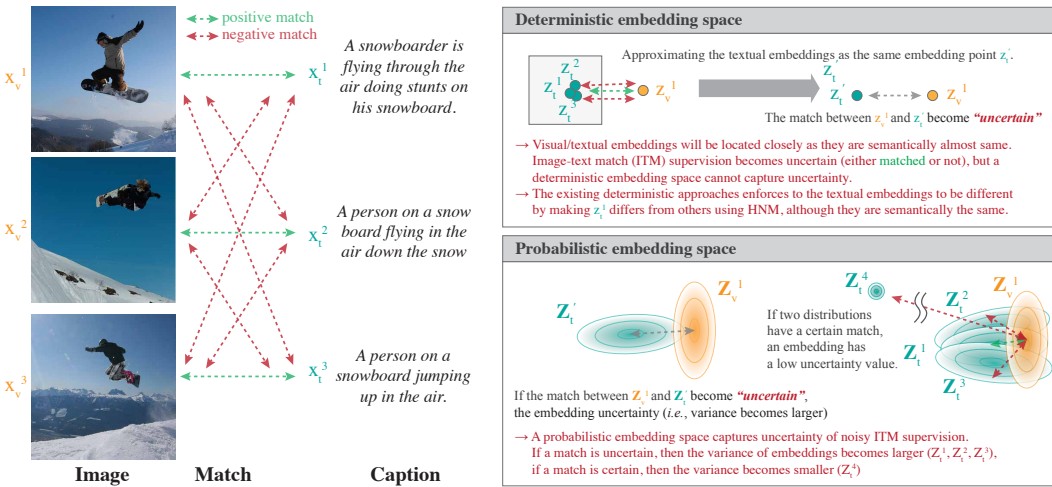

Figure 1: **Inherent ambiguity of ITM.** We assume that the deterministic textual embeddings are mapped to the same point $z'_t$, *i.e.*, $z^1_t \approx z^2_t \approx z^3_t \approx z'_t$, as well as the probabilistic textual embeddings $\mathbf{Z}^1_t \approx \ldots \approx \mathbf{Z}'_t$.

a deterministic vector. Probabilistic embeddings have been introduced for many applications with inherent ambiguity, such as face understanding (Shi & Jain, 2019; Chang et al., 2020), 2D-to-3D pose estimation (Sun et al., 2020), speaker diarization (Silnova et al., 2020), video understanding (Park et al., 2022b), and composed image retrieval (Neculai et al., 2022). Especially, Chun et al. (2021) investigated the primitive probabilistic approach for ITM, PCME. Although PCME shows reasonable retrieval performances and interesting observations through uncertainty measures, PCME suffers from expensive computations due to Monte Carlo approximation and fast loss saturation under FNs.

Firstly, PCME needs expensive sampling operations for both training and inference. For example, if we randomly draw 7 samples for each input, computing the distance between two samples costs $O(7 \times 7)$. Furthermore, due to the sampling operation, PCME retrieval operation cannot be extended to large-scale efficient retrieval systems, such as FAISS (Johnson et al., 2019). This issue is solved by introducing a new closed-form sampled distance (CSD), and a new objective function based on the distance (§2.2). In addition, as the CSD consists of Euclidean distance and the relationship between variance embeddings, we can easily adapt approximated KNN (ANN) to PCME++ (§3.5). Experimental results show that the closed-form distance not only makes the operation efficient but also convergences to a better solution by computing an exact solution instead of an approximation.

In addition, this paper proposes two optimization techniques to improve PCME++ under abundant false negatives (FNs) by introducing two soft label strategies that allow less penalty to potential FN pairs in the dataset: assigning pseudo-positives (PP) for high-confident samples (§2.3) and mixed sample data augmentation (MSDA) for probabilistic matching (§2.4). Note that applying soft labels to PCME++ is straightforward because PCME++ is based on a pair-wise loss function.

PCME++ is evaluated on MS-COCO Caption (Chen et al., 2015) and its extended benchmarks CxC (Parekh et al., 2021) and ECCV Caption (Chun et al., 2022) with state-of-the-art ITM methods (§3.2). PCME++ consistently works better than the comparison methods on the COCO benchmark, especially when the backbone size grows. PCME++ is also evaluated on the noisy correspondence benchmark (Huang et al., 2021), indicating that our method is not only effective for the original task but also holds the potential to address the noisy correspondence problem. Furthermore, this paper shows that the textual uncertainty of PCME++ can be applied to a prompt-filtering for a zero-shot classification with a pre-trained model on large-scale VL datasets, demonstrating the versatility and scalability of our method for a wide range of applications (§3.5). Finally, the qualitative advantages of the learned uncertainty of PCME++ by capturing dataset uncertainty are shown in §3.4.

**Contributions.** This paper shows that FNs lead to inherent ambiguity for VL datasets (Section 2.1) and proposes to solve the problem by PCME++. The newly introduced probability distance, named CSD, is more suitable to probabilistic representations as shown in Section 2.2. This paper also proposes two soft label optimization techniques: PPs and MSDA in Section 2.3. As PCME++

uses a pair-wise loss function invariant to other samples in mini-batch, it is straightforward to apply the soft labels. The extensive studies on COCO Caption show the effectiveness of PCME++. Impressively, while we can take advantage of probabilistic representations, such as interpretability (Figure 3, Figure 4, C.3), PCME++ performed the best in all backbones, especially when scaling-up backbones or under strong noisy correspondence. Finally, the primitive results of applying PCME++ to uncertainty-based prompt-filtering for zero-shot classification are demonstrated.

## 2 IMPROVED PROBABILISTIC CROSS-MODAL EMBEDDINGS (PCME++)

### 2.1 PROBLEM DEFINITION: AMBIGUITY OF ITM DATASETS

Let $x_v$ and $x_t$ be the input image and caption, respectively. For each image text pair, a binary matching indicator $m_{vt} \in \{0, 1\}$ denotes whether $x_t$ describes $x_v$ well. This paper argues that the inherent multiplicity and the sparse annotations make $m_{vt}$ ambiguous. For example, as shown in Figure 1, $x_t^1$ (*"A person on a snowboard flying in the air down the snow"*) and $x_t^2$ (*"A person on a snowboard jumping up in the air."*) are semantically almost the same, hence we may assume that $x_t^1$ and $x_t^2$ are mapped to almost the same embedding point $z_t'$, *i.e.*, $f(x_t^1) \approx f(x_t^2) = z_t'$ if we have a proper mapping $f(\cdot)$ between the input space and the embedding space. In this case, if $x_t^1$ and $x_v^1$ are a positive match, $x_t^2$ and $x_v^1$ should be a positive match in the embedding space. However, because our dataset contains only sparse matching relationships (Chun et al., 2022; Parekh et al., 2021), $x_t^2$ and $x_v^1$ are a negative match. In other words, in the embedding space, the matching between $z_v^1$ and $z_t'$ ($\approx f(x_t^1) \approx f(x_t^2)$) becomes ambiguous (*i.e.*, it can be either positive or negative). As shown in Figure 1, a deterministic embedding space cannot capture the inherent uncertainty originated by the multiplicity and the sparse annotations. The existing deterministic approaches, therefore, rely on the Hardest Negative Mining (HNM) strategy (Faghri et al., 2018), selecting the closest pair as the only negative for computing a triplet loss. The HNM strategy enforces sparse positive pairs to be closer than other false negative (FN) pairs, resulting in a twisted embedding space that cannot capture the inherent uncertainty of VL datasets. We empirically show that the HNM strategy eventually converges to a suboptimal embedding space when the ambiguity intensifies, *i.e.*, under strong noisy correspondences (§3.2). In contrast, probabilistic embeddings can naturally mitigate the issue by capturing the ambiguity of $m_{vt}$ with a probability distribution (Kirchhof et al., 2023).

### 2.2 PROBABILISTIC CONTRASTIVE LEARNING

We first define a visual embedding and a text embedding of the given image $x_v$ and $x_t$ as normally distributed random variables, $\mathbf{Z}_v \sim \mathcal{N}(\mu_v, \Sigma_v)$ and $\mathbf{Z}_t \sim \mathcal{N}(\mu_t, \Sigma_t)$, respectively. For simplicity, we assume diagonal covariance matrices and simplify the notations as $\mathcal{N}(\mu_v, \sigma_v^2)$ and $\mathcal{N}(\mu_t, \sigma_t^2)$, where $\mu$ and $\sigma$ are $D$-dimensional vectors. As shown in Figure 1, our purpose is to learn probabilistic embeddings $\mathbf{Z}_v$ and $\mathbf{Z}_t$ satisfying the following properties: **(a)** there exists a proper probabilistic distance between $\mathbf{Z}_v$ and $\mathbf{Z}_t$. **(b)** if the match $m_{vt}$ is certain, then $\mathbf{Z}_v$ and $\mathbf{Z}_t$ have small variances. **(c)** if the match between $x_v$ and $x_t$ ($m_{vt}$) is ambiguous, then $\mathbf{Z}_v$ and $\mathbf{Z}_t$ have large variances.

We define closed-form sampled distance **(CSD)**, between two probabilistic embeddings $\mathbf{Z}_v$ and $\mathbf{Z}_t$:

$$d(\mathbf{Z}_v, \mathbf{Z}_t) = \mathbb{E}_{\mathbf{Z}_v, \mathbf{Z}_t} \|\mathbf{Z}_v - \mathbf{Z}_t\|_2^2 = \|\mu_v - \mu_t\|_2^2 + \|\sigma_v^2 + \sigma_t^2\|_1, \tag{1}$$

where $\|\cdot\|_p$ is a p-norm operation. To be self-contained, the full derivation of Equation (1) is provided in Appendix A.1. Equation (1) satisfies most of the properties of a metric function (*i.e.*, positivity, symmetry, and triangular inequality) except zero self-distance; $d(\mathbf{Z}, \mathbf{Z})$ is $2\|\sigma^2\|_1$, not zero. *I.e.*, Equation (1) satisfies the condition **(a)**. There are two ways to make $\mathbf{Z}_v$ and $\mathbf{Z}_t$ closer/further; making $\mu_v$ and $\mu_t$ closer/further, or making $\sigma_v$ and $\sigma_t$ smaller/larger. Hence, if we assume fixed $\mu_v$ and $\mu_t$, we have to decrease $\sigma_v$ and $\sigma_t$ to minimize $d(\mathbf{Z}_v, \mathbf{Z}_t)$; if $\mathbf{Z}_v$ and $\mathbf{Z}_t$ are a certain positive match (*i.e.*, $m_{vt} = 1$), then $\sigma_v$ and $\sigma_t$ will be collapsed to zero (*i.e.*, satisfying the condition **(b)**), and $d(\mathbf{Z}_v, \mathbf{Z}_t)$ will become Euclidean distance. On the other hand, if the match between $\mathbf{Z}_v$ and $\mathbf{Z}_t$ is ambiguous (*i.e.*, $m_{vt}$ can be either positive or negative), then $\sigma_v$ and $\sigma_t$ will not be collapsed to zero, for increasing $d(\mathbf{Z}_v, \mathbf{Z}_t)$ for the negative match case; $d(\mathbf{Z}_v, \mathbf{Z}_t)$ also satisfies the condition **(c)**.

CSD has a similar form to Wasserstein 2-distance (WD), $\inf_{\mathbf{Z}_v, \mathbf{Z}_t} \mathbb{E}_{\mathbf{Z}_v, \mathbf{Z}_t} \|\mathbf{Z}_v - \mathbf{Z}_t\|_2^2 = \|\mu_v - \mu_t\|_2^2 + \|\sigma_v - \sigma_t\|_2^2$, where WD includes the infimum operation. However, WD is not a proper probabilistic distance in the matching problem, especially WD cannot satisfy the condition **(b)**. Assume the

scenario when $\mu$ values are fixed again. In this case, $\sigma_v$ and $\sigma_t$ have no motivation to be decreased, but they are just enforced to have the same values. Hence, the learned $\sigma$ by WD cannot represent the sample certainty. Figure A.1 shows a 2-D toy scenario where CSD satisfies the proper uncertainty conditions while WD cannot. In the figure, red, yellow, and green dots are certain samples, and others are uncertain samples. The size of each dot denotes the intensity of the learned $\sigma$ values. Here, we observe that $\frac{\bar{\sigma}^2_{\text{uncertain}}}{\bar{\sigma}^2_{\text{certain}}}$, the average $\sigma^2$ value for uncertain/certain samples by CSD are 1.82, while we have 1.04 for WD. More details of the toy experiment are described in Appendix A.2.

CSD is also related to the matching probability (Oh et al., 2019) used by PCME (Chun et al., 2021), where the matching probability cannot be computed in a closed-form but should be computed by an expensive Monte-Carlo approximation. Empirically, PCME++ is 33% faster than PCME for this reason. Furthermore, the PCME loss gives more weight to samples that correctly predict the distance relationships. However, our dataset has abundant FNs (*e.g.*, COCO captions have $\times 8.47$ positive images than the "ground-truth" positive images (Chun et al., 2022)), which leads to wrong distance relationship supervision. Hence, PCME can suffer from the gradient saturation problem under abundant FNs. The comparison between CSD and matching probability is discussed in Appendix A.3.

Now, based on Equation (1), the probabilistic matching objective function is defined as NLL loss:

$$\mathcal{L}_{\text{match}} = -m_{vt} \log \text{sigmoid}(-a \cdot d(\mathbf{Z}_v, \mathbf{Z}_t) + b) - (1 - m_{vt}) \log \text{sigmoid}(a \cdot d(\mathbf{Z}_v, \mathbf{Z}_t) - b), \quad (2)$$

where $m_{vt} \in \{0, 1\}$ is the matching indicator between $v$ and $t$. $a$ and $b$ are learnable scalar values, following Chun et al. (2021). In practice, Equation (2) can be easily implemented by binary cross entropy (BCE) loss. We compute $\mathcal{L}_{\text{match}}$ for all pairs in the mini-batch as contrastive learning objectives, such as InfoNCE (Radford et al., 2021). The overview of the comparisons between our objective function, a standard triplet loss, and batch-wise contrastive loss are shown in Figure A.3.

To prevent the collapse of $\sigma$ (*i.e.*, $\sigma \to 0$), PCME++ employs Variational Information Bottleneck (VIB) loss (Alemi et al., 2017), $\mathcal{L}_{\text{VIB}}$, following Chun et al. (2021). As derived by Oh et al. (2019), $\mathcal{L}_{\text{VIB}}$ can be computed by the KL divergence between the learned distribution and $\mathcal{N}(0, I)$.

### 2.3 PSEUDO-POSITIVES (PP) FOR HANDLING NUMEROUS FALSE NEGATIVES

In practice, we use a small mini-batch (*e.g.*, 128), which does not guarantee that all confusing samples are observed for each iteration. To tackle the issue, PCME++ employs a simple pseudo-positive (PP) strategy: for a positive match $(v, t)$, $t'$ is a PP match with $t$ if $d(\mathbf{Z}_v, \mathbf{Z}_{t'}) \leq d(\mathbf{Z}_v, \mathbf{Z}_t)$. Using the PPs, we compute the PP matching loss $\mathcal{L}_{\text{pseudo-match}}$ using equation 2. The objective function becomes:

$$\mathcal{L}_{\text{match}} + \alpha \mathcal{L}_{\text{pseudo-match}} + \beta \mathcal{L}_{\text{VIB}}, \quad (3)$$

where $\alpha$ and $\beta$ are control parameters of PP matching loss and VIB loss. In the experiments, $\alpha = 0.1$ and $\beta = 0.0001$ are chosen (Appendix C.3). Pseudo-code for Equation (3) is shown in Appendix A.4.

### 2.4 MIXED SAMPLE DATA AUGMENTATION (MSDA) FOR PROBABILISTIC MATCHING

MSDA, such as Mixup (Zhang et al., 2018) or CutMix (Yun et al., 2019), shows not only great improvements in empirical performances but also shows good theoretical properties, such as generalization (Zhang et al., 2021a; Park et al., 2022a) or calibration (Zhang et al., 2021b). MSDA consists of two parts; input mixing (*i.e.*, a generative process to generate a new mixed sample) and label mixing (*i.e.*, modifying the supervision of the mixed sample). The intensity of the augmentation is controlled by $\lambda$, usually sampled from a pre-defined Beta distribution. Usually, it is not straightforward to apply MSDA to metric learning or contrastive learning because their losses are computed in a batch-dependent way (Figure A.3 (a) and (b)). On the other hand, as our objective function is computed in a pair-wise manner (Figure A.3 (c)), it is easier to apply MSDA to our objective function. More detailed discussion of why MSDA is not applicable to previous methods is in Appendix A.5.

There are two issues with designing MSDA for probabilistic matching. First, MSDA for the textual modality is not straightforward. Hence, PCME++ only mixes visual inputs using Mixup and CutMix. Second, we cannot directly mix labels because our scenario has no class label. Instead, we let $m_{vt}$ smooth in Equation (2), *i.e.*, $m_{vt} \in [0, 1]$. This approach controls the smooth label by mixing intensity $\lambda$ by setting $m_{vt} = \lambda$.

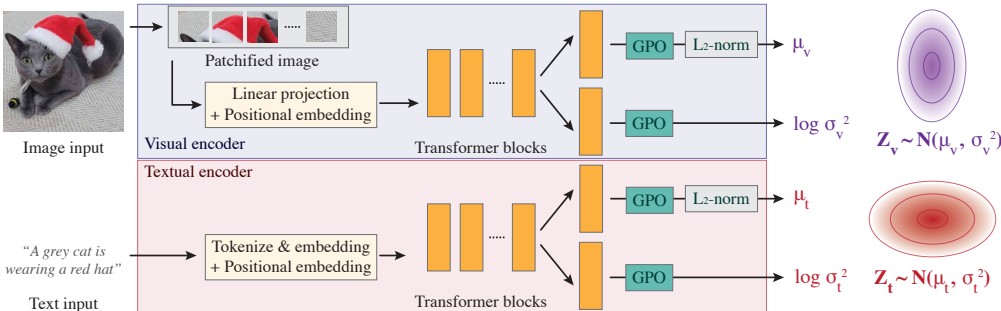

Figure 2: **Architecture overview.** We use the same visual and textual backbones as CLIP. Each modality encoder encodes $\ell_2$-normalized mean vector $\mu$ and the variance vector $\log \sigma^2$, followed by Generalized Pooling Operator (GPO) (Chen et al., 2021), to represent a normally distributed random variable $\mathbf{Z} \sim \mathcal{N}(\mu, \sigma^2)$.

The overview of the optimization procedure with PPs and MSDA is illustrated in Figure A.3 (d). In the experimental results, 25% of mini-batch images are mixed by sampling the mixing intensity $\lambda$ from Beta$(2, 2)$. For every mini-batch, Mixup or CutMix is randomly chosen for the mixing strategy. The empirical study shows that this strategy is slightly better than the widely-used batch-wise mixing strategy, *i.e.*, randomly mixing the whole mini-batch or using the original mini-batch (Appendix C.3).

## 2.5 ARCHITECTURE

PCME++ trains visual and textual encoders separately, such as visual semantic embeddings or CLIP. Each encoder has two heads, $\mu$ and $\log \sigma^2$ heads whose output vectors are $D$-dimensional. An input is mapped to a normal distribution parameterized by the output of $\mu$ and $\log \sigma^2$ heads.

PCME++ employs a Vision Transformer (ViT) (Dosovitskiy et al., 2021) as the visual backbone and a 12-layer 512-wide Transformer (Vaswani et al., 2017) as the textual backbone, following CLIP. PCME++ duplicates the last transformer layer for $\mu$ and $\log \sigma^2$ heads, *e.g.*, a textual backbone has a shared feature extractor with a 11-layer Transformer and $\mu$ and $\log \sigma^2$ are 1-layer Transformer blocks. The $\log \sigma^2$ head is randomly initialized, while the $\mu$ head is initialized as the same as the backbone initialization (*e.g.*, from a pre-trained model). We empirically observe that using more layers for $\log \sigma^2$ marginally improves the performances, but we set the number of layers for $\log \sigma^2$ head to 1 for computational efficiency. Finally, we employ Generalized Pooling Operator (GPO) for the feature aggregation with the same parameter setting of Chen et al. (2021). We observe that GPO brings both training stability and performance improvements. The model overview is illustrated in Figure 2.

## 3 EXPERIMENTS

### 3.1 EXPERIMENTAL PROTOCOL

Three evaluation benchmark datasets are used: COCO Caption (Chen et al., 2015), and its two extended benchmarks, ECCV Caption (EC) (Chun et al., 2022) and CxC (Parekh et al., 2021). Note that they have the same images and captions ($x_v$, $x_t$) but with different match annotations $m_{vt}$. The overview of each benchmark can be found in Appendix B.1. Following Chun et al. (2022), R@$k$ for all benchmarks and mAP@R and R-Precision (R-P) for EC are reported. The tables also show "RSUM", the summation of R@1, R@5, and R@10 for each modality retrieval on COCO 1K. The full results for each modality and R@$k$ results are in Appendix C.7. In the main paper, the averaged scores on each modality are reported. As we focus on the mitigation of the FN problem, we mainly focus on EC mAP@R, EC R-P, and COCO RSUM. Note that COCO and CxC have abundant FNs, hence their R@1 metrics could mislead to a wrong result (Musgrave et al., 2020; Chun et al., 2022).

**Comparison methods.** VSE$\infty$ (Chen et al., 2021) is based on a triplet loss and hardest negative mining (HNM) (Faghri et al., 2018). InfoNCE is the CLIP (Radford et al., 2021) pre-training objective. PCME (Chun et al., 2021) is a primitive probabilistic ITM model with sampling-based matching probability. This paper also evaluates two recent methods to tackle false negatives (FNs) of

ITM, DAA (Li et al., 2022a)[1] and P2RM (Wang et al., 2022), where both of them are an additional regularization method on top of the HNM triplet loss, *i.e.*, VSE∞. The optimization hyperparameters for all experiments are fixed, such as learning rate, based on VSE∞ ViT-B/32 validation rsum score. The hyperparameters of DAA and P2RM are searched in the same setting. As we initialize all models by CLIP pre-trained models, CLIP zero-shot (ZS) is also reported as a baseline. All models have the same visual and textual backbones, except probabilistic models; they have an additional $\log \sigma^2$ head (See Figure 2). All models are trained three times, and the average evaluation metric are reported.

**Training details and model selection.** PCME++ is initialized with the pre-trained CLIP model (Radford et al., 2021), while newly introduced modules, such as $\log \sigma^2$ head and GPO are randomly initialized. All models are trained for 25 epochs using AdamP optimizer (Heo et al., 2021). The training details can be found in Appendix B.2. The models are selected by the best validation RSUM following previous works. For the case when the model selection criterion is not possible, this paper also shows the SWA (Izmailov et al., 2018) results for ViT-B backbones (See Appendix B.3).

## 3.2 COCO ITM RESULTS

**Main results.** Table 1 shows the main comparison results of PCME++ and other ITM methods. For a fair comparison, PCME++ (SWA) is not compared to the other methods. We first observe that PCME++ consistently outperforms other methods in most evaluation metrics on different backbones, particularly on EC mAP@R, R-P, and RSUM. Second, we observe that the scale-up of PCME++ leads to consistent performance increases without hyperparameter tuning, while deterministic counterparts (*e.g.*, VSE∞ and InfoNCE) suffer from performance drops when scaling up from ViT-B/16 to ViT-L/14. As the backbone becomes more complex and larger, I presume that the backbone complexity of deterministic methods is sufficiently large to capture the noisy FNs in the dataset. Moreover, VSE∞ uses a triplet loss with the HNM, making the effect of FNs more significant (Chun et al., 2022). Meanwhile, the probabilistic methods are designed to handle many-to-many relationships with uncertainty representations (Section 2.1); hence, the effect of the FNs is saturated during training PCME and PCME++. Note that applying PP and MSDA to triplet loss (*e.g.*, VSE∞, DAA, and P2RM) is non-trivial because the triplet loss is not designed for taking smooth labels. More detailed discussions can be found in Appendix A.5. Third, we observe that the previously proposed methods to mitigate the FN problem, such as DAA and P2RM underperform than its baseline (VSE∞). It shows that the regularization-based deterministic methods cannot solve the FN problem due to the limitation of the deterministic embeddings. The full retrieval results for each modality and R@$k$ scores are reported in Appendix C.7.

**Noisy correspondence.** Table 2 shows the additional comparisons under noisy correspondence (NC), *i.e.*, by assuming that the training annotations are noisy. Following Huang et al. (2021), the image-text relationships are randomly shuffled with the probability of 20% and 50%. A specifically designed method for solving the NC problem, NCR (Huang et al., 2021) and DECL (Qin et al., 2022), are also compared with the comparison methods. Following Huang et al. (2021), the model selection criterion is also based on the clean validation rsum as the clean dataset scenario. Here, $\alpha$ for PP loss is set to 0.01 and 0.0 for 20% and 50% noise ratio because PPs can be incorrectly estimated under strong NC (Li et al., 2020). However, in Appendix C.2, we can observe that although using weaker PPs shows better by the best model selection by clean validation rsum, it eventually suffers from serve overfitting. This paper argues that the best model selection criterion based on the clean validation split should be reconsidered for future works to avoid a wrong conclusion. There are two findings in Table 2. First, the hardest negative mining-based triplet loss (VSE∞ and DAA) shows vulnerability on strong noisy correspondence, *e.g.*, 50%. Second, although the probabilistic methods, such as PCME and PCME++, are not designed for tackling NC, they successfully handle the scenario.

## 3.3 ABLATION STUDY

Table 3 shows that all the proposed techniques effectively improve probabilistic ITM. More detailed hyperparameter studies for each optimization (VIB, PP, and MSDA) are in Appendix C.3. Table 4 shows the impact of the probability distance on training objective (Equation (2)) by replacing

---

[1] Note that although DAA argues that it is an "approximation of probabilistic embedding", DAA is not a probabilistic method, but a deterministic regularization method based on text similarity.

Table 1: **COCO cross-modal retrieval performances.** Comparisons of ITM methods with various backbones in ECCV Caption, CxC and COCO Caption. "Prob?" denotes whether a method is a probabilistic method or not. Each number is the average between the image-to-text retrieval and text-to-image retrieval results, and is the average of three different runs. The full numbers and standard errors are in Appendix C.7. [†] denotes the re-evaluated results by the official checkpoints, otherwise, numbers are produced by our trained models.

| Backbone | Method | Prob? | ECCV Caption mAP@R | R-P | R@1 | CxC R@1 | 1K R@1 | COCO 5K R@1 | RSUM |
|---|---|---|---|---|---|---|---|---|---|
| ViT-B/32 (151M) | CLIP ZS[†] | ✘ | 26.8 | 36.9 | 67.1 | 42.0 | 59.5 | 40.3 | 471.9 |
| | VSE∞ | ✘ | 40.0 | 49.5 | **83.1** | **57.1** | **75.5** | **55.2** | 536.5 |
| | P2RM | ✘ | 39.0 | 48.7 | 82.0 | 53.6 | 73.3 | 51.7 | 530.2 |
| | DAA | ✘ | 39.2 | 49.0 | 82.0 | 54.8 | 73.6 | 52.9 | 530.9 |
| | InfoNCE | ✘ | 39.0 | 48.7 | 81.7 | 54.9 | 74.0 | 53.0 | 532.6 |
| | PCME | ✔ | 39.1 | 48.9 | 81.4 | 54.7 | 73.8 | 53.0 | 532.0 |
| | PCME++ ($\mu$ only) | ✘ | 39.5 | 49.1 | 82.7 | 57.0 | 75.3 | 55.2 | 536.2 |
| | PCME++ | ✔ | **40.1** | **49.7** | **83.1** | 56.8 | 75.4 | 55.1 | **537.0** |
| | PCME++ (SWA) | ✔ | 40.2 | 49.8 | 82.9 | 56.8 | 75.5 | 55.2 | 537.3 |
| ViT-B/16 (150M) | CLIP ZS[†] | ✘ | 29.3 | 39.0 | 71.1 | 44.3 | 62.0 | 42.7 | 481.0 |
| | VSE∞ | ✘ | 41.7 | 50.6 | 86.3 | 62.3 | 79.1 | 60.7 | 547.2 |
| | P2RM | ✘ | 39.7 | 49.5 | 80.7 | 54.2 | 73.8 | 52.5 | 532.7 |
| | DAA | ✘ | 20.7 | 30.6 | 50.2 | 25.4 | 43.7 | 23.4 | 410.2 |
| | InfoNCE | ✘ | 41.1 | 50.4 | 84.8 | 60.9 | 78.3 | 59.3 | 545.5 |
| | PCME | ✔ | 41.0 | 50.3 | 84.3 | 59.9 | 77.8 | 58.2 | 544.2 |
| | PCME++ ($\mu$ only) | ✘ | 41.2 | 50.4 | 85.7 | 62.5 | **79.3** | **61.0** | **548.0** |
| | PCME++ | ✔ | **42.1** | **51.2** | **86.5** | **62.6** | **79.3** | **61.1** | **548.0** |
| | PCME++ (SWA) | ✔ | 42.2 | 51.2 | 86.6 | 62.9 | 79.6 | 61.3 | 548.5 |
| ViT-L/14 (428M) | CLIP ZS[†] | ✘ | 28.0 | 37.8 | 72.2 | 48.1 | 64.8 | 46.4 | 491.6 |
| | VSE∞ | ✘ | 20.2 | 31.5 | 46.2 | 24.3 | 44.5 | 22.7 | 424.3 |
| | InfoNCE | ✘ | 35.6 | 45.8 | 75.6 | 48.0 | 69.5 | 45.9 | 520.6 |
| | PCME | ✔ | 41.2 | 50.3 | 86.0 | 63.4 | 80.3 | 61.9 | 550.4 |
| | PCME++ | ✔ | **42.1** | **50.8** | **88.8** | **65.9** | **81.8** | **64.3** | **554.7** |

Table 2: **COCO noisy correspondence.** Noisy correspondence results using the ViT-B/32 backbone (except NCR) are shown. NCR scores are re-evaluated by the official weights. As DECL does not provide the official weights, we report the scores from the paper. Noise ratio 0% is the same as Table 1.

| Noise ratio | Method | ECCV Caption mAP@R | R-P | R@1 | CxC R@1 | 1K R@1 | COCO 5K R@1 | RSUM |
|---|---|---|---|---|---|---|---|---|
| 20% | VSE∞ | 37.0 | 46.3 | 79.7 | **53.6** | **72.0** | **51.8** | 518.6 |
| | DAA | 6.7 | 12.5 | 18.5 | 7.0 | 15.3 | 6.0 | 212.8 |
| | InfoNCE | 35.9 | 46.3 | 76.1 | 47.8 | 68.2 | 45.8 | 514.6 |
| | PCME | 37.6 | **47.6** | 79.2 | 50.6 | 70.3 | 48.7 | 520.7 |
| | PCME++ (ours) | **37.7** | **47.6** | **80.0** | 52.2 | 71.6 | 50.4 | **524.6** |
| | NCR[†] | 35.9 | 46.0 | 78.0 | 50.6 | 70.1 | 48.8 | 518.6 |
| | DECL[†] | - | - | - | - | 69.6 | 49.4 | 518.2 |
| 50% | VSE∞ | 18.0 | 28.5 | 43.7 | 20.7 | 39.2 | 19.1 | 394.1 |
| | DAA | 0.3 | 0.8 | 1.0 | 0.3 | 0.8 | 0.2 | 20.9 |
| | InfoNCE | 33.6 | 44.1 | 73.0 | 43.5 | 64.0 | 41.4 | 499.5 |
| | PCME | 35.2 | 45.5 | 75.7 | 46.3 | 66.6 | 44.4 | 508.0 |
| | PCME++ (ours) | **35.7** | **45.8** | **76.3** | **47.4** | 67.6 | 45.5 | 511.0 |
| | NCR[†] | 34.0 | 44.3 | 75.1 | 47.3 | 66.8 | 45.5 | 508.5 |
| | DECL[†] | - | - | - | - | **68.3** | **46.8** | **513.5** |

$d(\mathbf{Z}_v, \mathbf{Z}_t)$. For a fair comparison, all newly proposed optimization techniques except VIB are omitted. As we already observed in Figure A.1, we confirm that Wasserstein distance is not a proper uncertainty estimate as a training objective. Also, the table shows that PCME++ outperforms PCME in all metrics. I presume it is because the matching probability is an approximated value by Monte Carlo approximation; therefore, the distance value will have an approximation gap.

More parameter studies for Table 3 and architecture study can be found in Appendix C.3

Table 3: **Effect of optimization methods.** Ablation study on VIB, Pseudo-Positives (PP) and Mixed Sample Data Augmentation (MSDA) with a ViT-B/32 backbone are shown.

| | | | ECCV Caption | | | CxC | | COCO | | |
|---|---|---|---|---|---|---|---|---|---|---|
| VIB | PP | MSDA | mAP@R | R-P | R@1 | R@1 | 1K R@1 | 5K R@1 | RSUM |
| ✗ | ✗ | ✗ | 38.9 | 48.6 | 82.2 | 56.7 | 75.2 | 54.9 | 535.9 |
| ✔ | ✗ | ✗ | 39.3 | 49.0 | 83.1 | 56.1 | 74.5 | 54.3 | 534.5 |
| ✗ | ✔ | ✗ | 39.0 | 48.6 | 82.7 | **56.8** | 75.2 | 55.0 | 536.0 |
| ✗ | ✗ | ✔ | 39.0 | 48.6 | 82.1 | 56.4 | 74.9 | 54.6 | 535.5 |
| ✔ | ✔ | ✗ | 39.6 | 49.2 | 82.6 | 56.3 | 74.8 | 54.5 | 534.8 |
| ✔ | ✔ | ✔ | **40.1** | **49.7** | **83.1** | **56.8** | **75.4** | **55.1** | **537.0** |

Table 4: **Effect of probability distance on training objective.** Results on ViT-B/32 backbone with VIB loss.

| | ECCV Caption | | | CxC | | COCO | |
|---|---|---|---|---|---|---|---|
| Probability distance | mAP@R | R-P | R@1 | R@1 | 1K R@1 | 5K R@1 | RSUM |
| KL Divergence | 0.0 | 0.1 | 0.0 | 0.0 | 0.1 | 0.0 | 3.2 |
| JS Divergence | 0.0 | 0.1 | 0.1 | 0.0 | 0.1 | 0.0 | 3.2 |
| Wasserstein 2-distance | 1.9 | 4.2 | 6.7 | 3.8 | 9.0 | 3.5 | 121.1 |
| Expected Likelihood Kernel | 36.5 | 46.0 | 82.0 | **56.3** | 74.1 | **54.7** | 529.0 |
| Bhattacharyya distance | **39.3** | 48.9 | 80.7 | 53.7 | 72.5 | 51.8 | 524.9 |
| Match probability by PCME | 39.1 | 48.9 | 81.4 | 54.7 | 73.8 | 53.0 | 532.0 |
| Proposed (Equation (1)) | **39.3** | **49.0** | **83.1** | 56.1 | **74.5** | 54.3 | **534.5** |

## 3.4 Uncertainty analysis

From Equation (1), we can define the data uncertainty as $\|\sigma^2\|_1$, *i.e.*, the summation of the variance. Based on the data uncertainty, Figure 3 shows how the uncertainty captures the ambiguity of datasets. The average COCO 1K R@1s for each modality in each of the 10 uncertainty bins are reported in the figure. We observe that by the uncertainty increased, COCO R@1 (the same distribution as the training dataset) is decreased. The results support that the learned uncertainty by PCME++ can capture the inherent ambiguity of the matching annotations.

Figure C.3 shows examples of uncertain images and captions, and their retrieved items. The figure shows that data that can be matched with more samples have higher uncertainty values. As shown in the figure, the retrieved items for uncertain inputs are highly plausible even though the retrieved items are not in the COCO ground truth. Section 3.5 and D discuss more benefits of uncertainty-aware learning and the learned uncertainty.

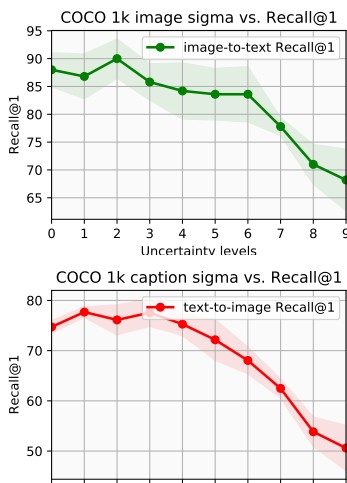

Figure 3: $\|\sigma^2\|_1$ **vs. R@1.**

Figure 4 shows the t-SNE visualization of ViT-B/32 PCME++ and VSE∞ in Table 1. As shown in Figure 4, the PCME++ embedding space captures the uncertain captions by enlarging their variances; thereby, each uncertain caption embedding can represent the uncertainty caused by the multiplicity of the dataset. On the other hand, due to the HNM strategy, the learned embedding space by VSE∞ (Figure 4 right) cannot correctly map three different images and captions with multiplicity. We also observe the same phenomenon in Figure A.1; it shows that HNM will ruin the embedding space. More details are in Appendix A.2. We also provide more detailed explanation of Figure 4 in Appendix C.4.

## 3.5 More applications

**Large-scale retrieval system.** Lack of scalability is a common drawback of probabilistic retrieval systems, *i.e.*, it is difficult to apply probabilistic embeddings on a large-scale retrieval system with a billion-scale index. As our CSD (Equation (1)) is the sum of Euclidean distance of $\mu$ and the intensity of $\sigma^2$ of each input, we can easily and efficiently combine PCME++ and approximated KNN (ANN).

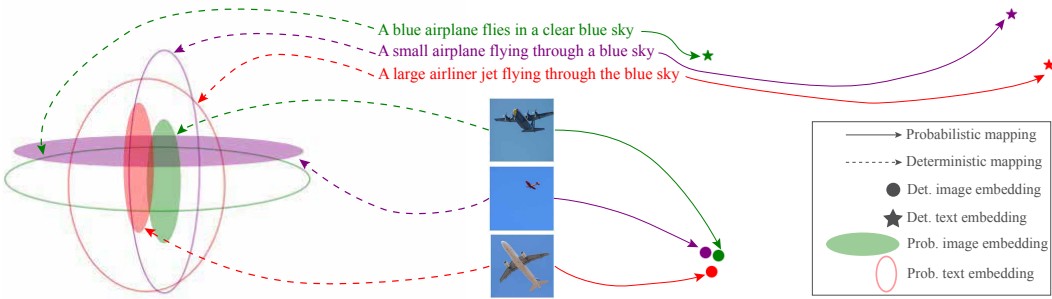

Figure 4: **2D t-SNE visualization of learned embeddings by PCME++ and VSE∞.** The area of probabilistic embeddings denotes the uncertainty of each embedding, *i.e.*, a more uncertain sample has a larger area.

Appendix C.5 describes the details of the modified ANN, and the comparisons of diverse retrieval strategies. CSD is not only stronger than other probability distances but also more practical.

**Uncertainty-based prompt-filtering.** Zero-shot (ZS) classification is the task of predicting an unseen class during training. Usually, ZS classification is done by converting class information as a text sentence (*e.g.*, "a photo of a cat") and mapping into a shared embedding space with the input. For image ZS classification tasks, large-scale ITM pre-training, such as CLIP, has become a standard approach. Despite their usability and generalizability, ZS needs hand-crafted prompt engineering for converting class information into proper text sentences.

Table 5: **ImageNet (IN) Zero-shot (ZS).**

| Model | Prompts | Top-1 Acc |
|---|---|---|
| InfoNCE | *"A photo of* $\{\cdot\}$*"* | 31.85 |
| | All 80 prompts | 35.50 |
| PCME++ | *"A photo of* $\{\cdot\}$*"* | 30.43 |
| | All 80 prompts | 34.22 |
| | Top-K certain prompts | 34.22 |
| | Best top-K for each class | **41.82** |

For example, Radford et al. (2021) showed that taking the average of 80 different context prompts improves ImageNet top-1 ZS accuracy by 3.5% over a single prompt (*"a photo of* $\{\cdot\}$*"*). However, designing the best-performing prompts for every novel task is time-consuming.

This paper investigates the potential of PCME++ for automatic prompt engineering using the learned text uncertainty: First, the uncertainties of prompts for each class are computed, (*e.g.*, *"a photo of a cat"*, *"a photo of many cat"*, ...), and then the most uncertain text prompts are discarded; this process is computed directly on the ImageNet validation set (*i.e.*, it is not a true "ZS". See Appendix C.6 for the more detailed discussion). Table 5 shows a study on the proposed simple automatic prompt-filtering. For the experiment, ViT-B/16 models using InfoNCE loss and PCME++ are trained on CC3M Sharma et al. (2018), 12M (Changpinyo et al., 2021) and RedCaps (Desai et al., 2021), for 32 epochs with 1024 batch size. "Top-K certain prompts" denotes that every class uses the same top-K for the filtering, and "Best top-K for each class" denotes the best top-K for each class are chosen, *e.g.*, "terrace" needs all 80 prompts, while "pencil case" only needs Top-1 certain prompt while other uncertain 79 prompts are discarded. More examples of the selected prompts are shown in Figure C.4. With this simple strategy, the ZS performance is increased with a significant gap (30.43 → 41.82). The full description of the ZS experiment is provided in Appendix C.6.

## 4  CONCLUSION

This paper brings attention to the importance of the FNs in ITM training datasets; a deterministic method will fail to capture the inherent ambiguity of the dataset, especially for a large backbone, such as ViT-L. This paper addresses the inherent ambiguity of ITM tasks by probabilistic embedding with a novel closed-form probability distance and a new matching objective for efficiency and effectiveness. The proposed method, PCME++, is further enhanced by employing a pseudo-positive strategy and a mixed sample data augmentation strategy, thanks to the pair-wise loss function design. Experimental results demonstrate the extensibility of PCME++ to various applications, such as image-text cross-modal retrieval, mitigating noisy correspondences, automatic prompt-filtering for zero-shot classification, and understanding the inherent ambiguity of a dataset. Beyond performance, the learned uncertainty by PCME++ shows high interpretability of the datasets as well as the controllability by the users when the rejection of the retrieved items is required.

ACKNOWLEDGEMENT

I would like to thank NAVER AI Lab colleagues for valuable discussions, including Sangdoo Yun, Wonjae Kim, Jiyoung Lee, Dongyoon Han, Byeongho Heo, Taekyung Kim, Song Park and Jung-Woo Ha. NAVER Smart Machine Learning (NSML) platform (Kim et al., 2018) is used for the experiments. I would like to express my sincere gratitude to my adorable cat, Seul Park, who served as the model for Figure 2, and my wife, Song Park, the delightful companion who brightened my research journey.

SOCIETAL IMPACT

This work aims to learn better image-text representations based on a probabilistic approach. As shown in the experiments, PCME++ has the potential to improve the interpretability and the controllability of learned representations by providing an additional degree of freedom to the users. Accordingly, PCME++ shares the potential impact of developing general image-text representations with better interpretability and controllability. For example, as shown by Radford et al. (2021), visual-textual representations trained on a large-scale web dataset often suffers from biases in the web; PCME++ can both mitigate or enhance the biases using its interpretability and controllability.

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

## APPENDIX

More additional materials are included here. More details of PCME++ are described in §A, including the full derivation of the closed-form probabilistic distance (§A.1), the toy experiments (§A.2), comparisons between PCME and PCME++ (§A.3), pseudo-code of PCME++ (§A.4), and why applying PPs and MSDA is non-trivial to other methods (§A.5). In experimental protocol details section (§B), the benchmark dataset details (§B.1), hyperparameter, resource details (§B.2) and SWA details (§B.3) are shown. Additional experimental results (§C), including comparisons with state-of-the-art (§C.1), the analysis of pseudo-positives (§C.2) the full ablation studies (§C.3), the t-SNE visualization details and related discussions (§C.4) the comparisons of different retrieval strategies (§C.5), automatic prompt-filtering experiments (§C.6), and the full experimental results and error bars (§C.7) are presented. Finally, discussions and limitations of PCME++ are in §D.

## A   METHOD DETAILS

### A.1   DERIVATION OF THE CLOSED-FORM PROBABILITY DISTANCE

In this subsection, the full derivation of Equation (1) is shown. We first show two simple well-known lemmas and conclude the full proof using them.

**Lemma 1.** *Let $X$ and $Y$ be independent normally distributed random variables where $X \sim \mathcal{N}(\mu_X, \Sigma_X)$ and $Y \sim \mathcal{N}(\mu_Y, \Sigma_Y)$. Then, the subtraction between $X$ and $Y$ is another normal distribution, i.e., $(X - Y) \sim \mathcal{N}(\mu_X - \mu_Y, \Sigma_X + \Sigma_Y)$.*

*Proof.* Let $\phi_X(u) = \exp(it^\top \mu_X - \frac{1}{2}t^\top \Sigma_X t)$ be a characteristic function of normally distributed random variable $X$. Using the fact that $-Y \sim \mathcal{N}(-\mu_Y, \Sigma_Y)$, we can compute the summation of $\phi_X(u)$ and $\phi_{-Y}(u)$ as follows:

$$\phi_{X-Y}(u) = \exp(it^\top \mu_X - \frac{1}{2}t^\top \Sigma_X t)\exp(-it^\top \mu_Y - \frac{1}{2}t^\top \Sigma_Y t) = \exp(it^\top(\mu_X - \mu_Y) - t^\top(\Sigma_X + \Sigma_Y)t). \tag{A.1}$$

Hence, $X - Y$ is another normal distribution, $\mathcal{N}(\mu_X - \mu_Y, \Sigma_X + \Sigma_Y)$. □

**Lemma 2.** *Let $X \sim \mathcal{N}(\mu, \Sigma)$. Then $\mathbb{E}\|X\|_2^2 = \|\mu\|_2^2 + tr(\Sigma)$.*

*Proof.* We first re-parameterize a random variable $X$ as $X = \mu + SZ$, where $S$ is the square root matrix of $\Sigma$, *i.e.*, $SS^\top = \Sigma$, and $Z$ is a standard normal distribution. Note that $S$ always exists because $\Sigma$ is a positive semi-definite by definition. Using $\mathbb{E}[Z] = 0$, the property of Frobenius norm $\|A\|_F^2 = tr(A)$ and the property of trace $tr(AB) = tr(BA)$, we have:

$$\begin{aligned} E\|X\|_2^2 &= \mathbb{E}_Z[\|\mu\|_2^2 + 2\mu SZ + \|Z^\top S^\top SZ\|_2^2] = \|\mu\|_2^2 + \mathbb{E}_Z\|Z^\top S^\top SZ\|_2^2 \\ &= \|\mu\|_2^2 + \mathbb{E}_Z tr(Z^\top S^\top SZ) = \|\mu\|_2^2 + tr(S^\top S \, \mathbb{E}_Z[ZZ^\top]) = \|\mu\|_2^2 + tr(\Sigma). \end{aligned} \tag{A.2}$$

□

**Proposition 1.** *Let $X$ and $Y$ be independent normally distributed random variables where $X \sim \mathcal{N}(\mu_X, \Sigma_X)$ and $Y \sim \mathcal{N}(\mu_Y, \Sigma_Y)$. Then we have $\mathbb{E}\|X - Y\| = \|\mu_X - \mu_Y\|_2^2 + tr(\Sigma_X + \Sigma_Y)$.*

*Proof.* By combining Lemma 1 and Lemma 2, the proof is completed. □

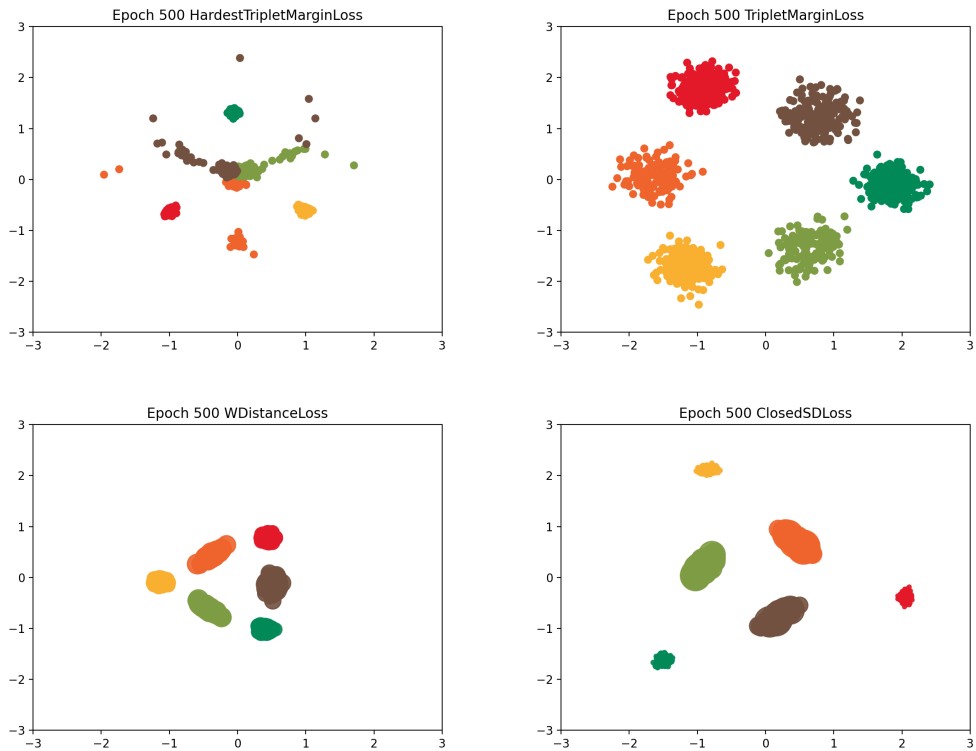

Figure A.1: **Toy results for HNM & SUM VSE, Wesserstien distance & PCME++.** The full animation can be found in https://naver-ai.github.io/pcmepp/.

## A.2   TOY EXPERIMENTS

In Section 2.2, a 2-D toy dataset is introduced for comparing various objective functions under inherent uncertainty. The toy dataset has three classes with "confusing samples" between classes, *i.e.*, a confusing sample randomly can be either class A or class B. The number of confusing samples are 30% of the total data points. To synthesize the samples, a centroid is randomly chosen for each class. Using the centroid, each sample is randomly drawn from $\mu + 0.1 \times \mathcal{N}(0, I)$. 500 samples are drawn for each class and 150 samples of them are chosen as "confusing samples", *i.e.*, there are 1500 samples with 1050 certain samples and 450 confusing samples. Then, $\log \sigma$ of each sample is randomly drawn from $\mathcal{U}(-1.5, 1.5)$ where $\mathcal{U}$ is a uniform distribution. In summary, the dataset has 350 confident samples for class 1, 2 and 3; 150 confusing samples for class (1, 2), (2, 3) and (3, 1).

To show the effects of different probabilistic distances, the samples are directly updated by the objective function Equation (2) with different metrics, *i.e.*, a sample $(\mu, \sigma)$ is directly updated by Equation (2). The dataset is directly optimized using Adam optimizer (Kingma & Ba, 2015) with learning 0.02 during 500 epochs. The mini-bath size is set to 128. The same loss function with PCME++ (*i.e.*, Equation (2)) is employed while the probabilistic distance $d(\mathbf{Z}_v, \mathbf{Z}_t)$ is chosen from either CSD (Equation (1)) or Wasserstein distance. VSE models are also trained with the hardest negative mining and without the negative mining (*i.e.*, using the summation of triplet distances). The last snapshots of each model are shown in Figure A.1 The animated learning progress of each method can be found in https://naver-ai.github.io/pcmepp/.

As described in Section 2.2, we desire the learned uncertainty can capture the data uncertainty, *i.e.*, we expect that certain samples have small $\sigma^2$, while uncertain samples have large $\sigma^2$. After training, we observe that the average $\sigma^2$ for certain samples and uncertain samples by PCME++ are 1.68 and 3.05, respectively. On the other hand, Wasserstein distance shows 2.69 and 2.80, respectively. The result and other experiments on large-scale datasets (Section 3.3) support that PCME++ is a proper probability distribution to capture uncertainty, while Wasserstein is not.

## A.3 COMPARISONS WITH PCME AND PCME++ OBJECTIVE FUNCTIONS

As both PCME and PCME++ aim to learn probabilistic embeddings, they share the nature of the probabilistic embeddings: we train an encoder that maps an input to a mean vector and a variance vector and train the encoder by maximizing the negative loglikelihood (NLL) using the extracted distributions, namely $\min -\sum \log p(m|x_v, x_t)$. PCME and PCME++ optimize different NLLs where PCME is based on "matching probability" and PCME++ is based on binary cross-entropy.

We recall the definition of matching probability of PCME (Chun et al., 2021):

$$p(m|x_v, x_t) = \mathbb{E}_{Z_v, Z_t} \, \text{sigmoid}(-a\|Z_v - Z_t\|_2 + b) \approx \frac{1}{J^2} \sum_{z_v, z_t} \text{sigmoid}(-a\|z_v - z_t\|_2 + b), \quad \text{(A.3)}$$

where $J$ is the number of samples $z_v$ and $z_t$. PCME directly optimized the negative log-likelihood:

$$m_{vt} \log \sum_{z_v, z_t} \text{sigmoid}\left(-a\|z_v - z_t\|_2 + b\right) + (1 - m_{vt}) \log \sum_{z_v, z_t} \text{sigmoid}\left(a\|z_v + z_t\|_2 + b\right) \quad \text{(A.4)}$$

On the other hand, PCME++ optimizes the NLL using binary cross-entropy loss (Equation (2)):

$$p(m|x_v, x_t) = \text{sigmoid}(-a\mathbb{E}_{Z_v, Z_t}\|Z_v - Z_t\|^2 + b).$$

Equation (A.4) and Equation (2) share a similar formulation, but the position of the expectation is different. As the expectation is located at the outside of $\text{sigmoid}$, Equation (A.3) cannot be computed in a closed-form solution, but our distance can. PCME++ has two benefits over PCME: (1) our form has a closed-form solution, while PCME cannot. (2) our form can be naturally adopted into the binary cross entropy loss function, which is known to be stable and perform well in large-scale training (Wightman et al., 2021).

Furthermore, as pointed out in Section 1, The computational cost of PCME depends on the number of MC samples $J$, because it needs to compute $O(J^2)$ pairwise distances between all samples. When we use the same setting of the paper ($J = 8$), we observe that PCME++ 25 epoch training takes 106,311 secs (1 day and 5 hours), while PCME 25 epoch training takes 141,694 secs (1 day and 15 hours) on a single V100 GPU. Overall, PCME needs 33% more training time compared to PCME++. Note that if we increase the sampling size $J$, the gap becomes larger. Another issue of the PCME sampling is that we need more memory size when computing the Monte Carlo approximation for a larger sampling size. Overall, PCME needs more forward time than PCME++ (33% more), and more memory size than PCME++ (on average, 18% more, but it is not a rigorous comparison because PCME has higher peak memory usage).

## A.4 PCME++ PSEUDO-CODE

Figure A.2 shows the PyTorch style pseudo-code of PCME++. Note that $\mu$ and $\sigma$ are extracted from the augmented inputs, such as MSDA (Section 2.4) and SizeAugment (Chen et al., 2021).

## A.5 WHY IS IT NON-TRIVIAL TO APPLY MSDA TO PREVIOUS METHODS?

Although this paper does not propose a new MSDA, the contribution of this paper lies in applying MSDA to the relational datasets. For example, applying MSDA to classification is straightforward because the mixed sample does not affect the other samples in the mini-batch. However, in the relational training objectives, such as triplet loss or contrastive loss, a mixed sample affects the other samples in the batch as well. Especially, the triplet loss is impossible to handle MSDA, because the core concept of MSDA is the smooth label, but the triplet loss cannot handle smooth label, because it has to construct a triplet of the selected sample, the positive sample, and the negative sample. It is non-trivial to define positive and negative samples when the label is smoothed (See Figure A.3 a). For example, assume that we set a match annotation of $v_a, t_a$ to 0.6 from 0.0. In this case, it is non-trivial to build triplets using this annotation. Moreover, if we introduce mixed samples and mixed labels, the problem becomes more complex. How can we handle $v_{a,b}$ (a mixed image of $x_a$ and $x_b$) and $t_a$, or $v_{b,a}$ and $t_b$ using a triplet relationship, or a pairwise relationship? Therefore, it is non-trivial to apply PPs and MSDA for the triplet-based methods.

Similarly, a batch-wise contrastive loss, such as InfoNCE, is also a little bit tricky to control the effect of smooth labels (See Figure A.3 b) because the mixed samples are combined in the denominator

```python
def compute_loss(v_mu, v_sig, t_mu, t_sig, matched):
    """v_mu, v_sig: mean and variance for (mixed) images (N by D)
       t_mu, t_sig: mean and variance for captions (M by D)
       matched: denoting (i, j) image, caption pair is matched.
                values are between 0 and 1 (N by M)"""
    # compute a closed-form distance
    mu_dist = ((v_mu.unsqueeze(1) - t_mu.unsqueeze(0)) ** 2).sum(-1)
    sigma_dist = ((v_sig.unsqueeze(1) + t_sig.unsqueeze(0))).sum(-1)

    # a, b: a learnable affine transform
    logits = -a *  (mu_dist + sigma_dist) + b

    # match loss can be easily computed by BCE loss
    match_loss = BCE(logits, matched)

    # compute pseudo-positive (pp) match loss
    gt_labels, gt_indices = torch.max(matched, dim=1)
    gt_vals = logits[:, gt_indices].diag()
    pseudo_gt_indices = (logits >= gt_vals)
    pp_matched = (gt_labels.unsqueeze(1) * (pseudo_gt_indices))
    matched[pseudo_gt_indices] = pp_matched[pseudo_gt_indices]
    pp_match_loss = BCE(logits, matched)

    # compute VIB loss
    v_vib = -0.5 * (1 + torch.log(v_sig) - v_mu ** 2 - v_sig).mean()
    t_vib = -0.5 * (1 + torch.log(t_sig) - t_mu ** 2 - t_sig).mean()
    vib_loss = v_vib + t_vib

    # final loss, alpha and beta are hyperparemeters
    return match_loss + alpha * pp_match_loss + beta * vib_loss
```

Figure A.2: **PyTorch pseudo-code of PCME++.** Here, `v_sig` and `t_sig` are computed by taking an exponential to the output of $\log \sigma^2$ heads. `BCE` denotes a binary cross-entropy function.

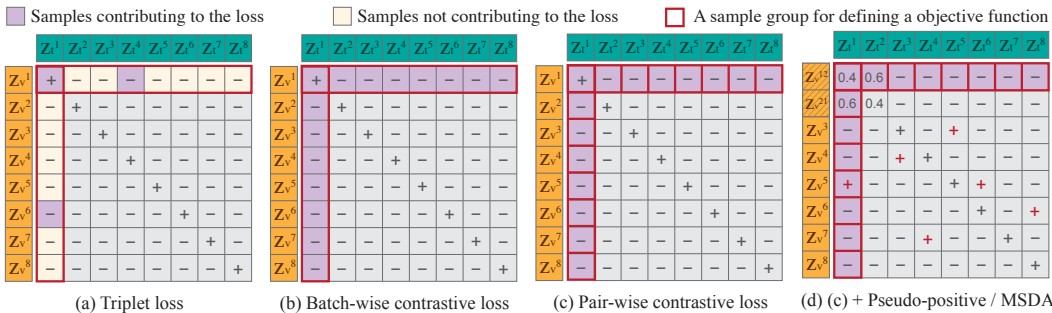

(a) Triplet loss     (b) Batch-wise contrastive loss     (c) Pair-wise contrastive loss     (d) (c) + Pseudo-positive / MSDA

Figure A.3: **Comparisons of different objective functions.** For given $i$-th visual embeddings $\mathbf{z}_v^i$ and $j$-th textual embedding $\mathbf{z}_t^j$, we illustrate how each sample contributes to different loss functions. (a) Only two image-caption pairs contributed to the loss in each row/column for triplet loss: $\mathcal{L}_v^i = [\|z_v^i - z_t^i\| - \|z_v^i - z_t^n\| + \alpha]_+$, where $\mathcal{L}_v^i$ is the loss value of $i$-th visual feature $z_v^i$, and $\mathcal{L}_t^j$ is defined similarly. (b) Batch-wise contrastive loss, such as InfoNCE, is defined for each row/column: $\mathcal{L}_v^i = \mathrm{CE}\left(\frac{\exp(-\|z_v^i - z_t^i\|)}{\sum_j \exp(-\|z_v^i - z_t^j\|)}, i\right)$, where CE denotes cross-entropy loss. Namely, the entire textual features are used to compute the loss value of $i$-th visual feature. (c) Pair-wise contrastive loss, such as PCME++, is defined for each image-caption pair: $\mathcal{L}_v^{ij} = \mathrm{BCE}(\mathrm{sigmoid}(d(\mathbf{Z}_v, \mathbf{Z}_t)), \mathbb{I}_{i=j})$, where $d(\cdot, \cdot)$ is CSD (Equation (1)) and $\mathbb{I}$ is an indicator function. Hence, our loss is computed multiple times for each row/column: $\mathcal{L}^{\mathrm{pairwise}} = \sum_{i,j} \mathcal{L}^{ij}$. On the other hand, (a) and (b) are computed by $\mathcal{L}^{\mathrm{others}} = \sum_i \mathcal{L}_v^i + \sum_j \mathcal{L}_t^j$. (d) As our loss is computed pair-wise, it is straightforward to apply pseudo-positives (PPs) or mixed sample data augmentation (MSDA), while it is not trivial to apply PP and MSDA to other methods as described in Appendix A.5.

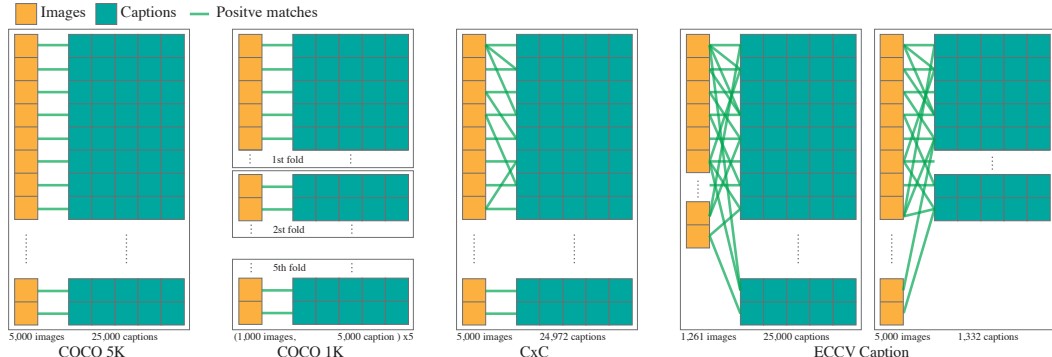

Figure B.1: **Difference between COCO 5K, 1K (Chen et al., 2015), CxC (Parekh et al., 2021) and ECCV Caption (Chun et al., 2022).** All matches not illustrated in the image are negative. ECCV Caption has separated query sets for each modality, while other datasets use the same images and captions for both query and gallery.

Table B.1: **Hyperparameter details**

| Method | ViT B/32, B/16, L/14 COCO | ViT B/16 CC3M, 12M and RedCaps |
|---|---|---|
| Epochs | 25 | 32 |
| Batch size | 128 | 1,024 |
| Optimizer | AdamP | AdamW |
| Initial learning rate | 0.0005 | 0.0005 |
| LR scheduling | Step | linear warmup and cosine |
| Layer-wise LR decay | 0.7 | - |
| Visual backbone LR decay | 0.01 | - |
| Textual backbone LR decay | 0.1 | - |
| $\beta_1, \beta_2, \varepsilon$ | $0.9, 0.999, 10^{-8}$ | $0.9, 0.98, 10^{-6}$ |
| Weight decay | 0.0001 | 0.2 |
| VIB $\beta$ | 0.0001 | $10^{-6}$ |
| PP $\alpha$ | 0.1 | 0 |
| MSDA CutMix/Mixup $\lambda$, mix ratio | 2/2/25% | -/-/0% |
| Size Augment | ✔ | ✘ |
| Embedding dimension | 1024 | 512 |
| Initial $a$ and $b$ | 5/5 | 1/1 |
| Resources and training hours | ViT B/32 1 V100 (38 hours) ViT B/16 1 V100 (75 hours) ViT L/14 8 V100 (62 hours) | 8 V100 (17 hours) |

term of Softmax. On the other hand, the proposed pairwise contrastive loss can directly apply smooth labels because each loss computation is invariant to the other samples, but only the given pairs affect the loss computation as shown in Figure A.3 c.

One of the technical contributions of this paper is allowing smooth labels and mixed samples by designing a pairwise loss that is not affected by the other data samples. As shown in Figure A.3 d, each loss computation of PCME++ is independent of the other pairs, while triplet loss or batch-wise contrastive loss is dependent on the relationships of other pairs.

# B   EXPERIMENTAL PROTOCOL DETAILS

## B.1   MORE DETAILS OF BENCHMARK DATASETS

PCME++ is evaluated on MS-COCO Caption (Chen et al., 2015), a widely used ITM benchmark, containing 123,287 images from MS-COCO (Lin et al., 2014) and five human-annotated captions per image. 113,287/5,000/5,000 images are used for training/validation/testing (Karpathy & Fei-Fei, 2015). Although Recall@$k$ (R@$k$) is a common evaluation metric in COCO Caption, as Musgrave

et al. (2020) showed, R@$k$ is often insufficient to measure retrieval performances. Furthermore, recent studies (Parekh et al., 2021; Chun et al., 2022) observed that many COCO Caption negatives are actually positives; *e.g.*, Chun et al. (2022) showed that 88.2% and 72.1% positive images and captions are annotated as negative in COCO. In other words, COCO R@$k$, relying on the noisy COCO annotation $m_{vt}$, is not fully reliable.

To mitigate the problem of R@$k$ evaluation, two extended benchmarks, ECCV Caption (EC) (Chun et al., 2022) and CxC (Parekh et al., 2021), are employed for the test split. Both datasets are validated by human annotators; EC contains more plentiful positives than CxC but its queries are the subset of the original COCO Caption; CxC has fewer positives than EC, but its annotations cover the whole COCO test split, and the annotations are less noisy. Note that the original COCO Caption, EC, and CxC have the same images and captions $(x_v, x_t)$ but with different match annotations $m_{vt}$.

Figure B.1 illustrates the differences between evaluation datasets. Note that all evaluation benchmarks use the same training dataset described in §3.1. The COCO Caption evaluation split consists of 5,000 images and 25,000 captions. **COCO 5K** uses the full 5,000 images and 25,000 captions where each image has five positive captions and each caption only has one positive image. For evaluation, COCO 5K measures image-to-text retrieval performances by setting 5,000 images as queries and 25,000 captions as galleries, while text-to-image retrieval performances are measured in the opposite way. **COCO 1K** uses the same positive relationships as COCO 5K, but COCO 1K uses the subset of COCO 5K, *i.e.*, there are 1,000 images and their corresponding 5,000 captions for COCO 1K split. COCO 1K measures the performances by taking an average of five different splits.

CxC (Parekh et al., 2021) and ECCV Caption (Chun et al., 2022) use the same images and captions of COCO 1K/5K, but with more positive annotations. CxC uses the entire images and valid 24,972 captions among 25,000 captions (by omitting *"I cannot see any image"* captions). CxC has more positive annotations than COCO, but there are still many missing positives in CxC because their approach is mostly focused on text similarity, not image-text similarity. On the other hand, ECCV Caption is designed for handling false negatives of image-text pairs. ECCV Caption uses the subset of images and captions for the queries, but their retrieval database is the full dataset, *i.e.*, when performing image-to-text retrieval, the number of query images is 1,261 and the number of gallery captions are 25,000; for text-to-image retrieval, the number of query texts is 1,332 and the number of gallery images is 5,000.

As discussed by Musgrave et al. (2020) and Chun et al. (2022), Recall@K is not an informative metric for measuring retrieval performances in terms of precision. Due to this reason, this paper reports mAP@R and R-Precision of ECCV Caption as the main comparison metrics.

## B.2 HYPERPARAMETER AND RESOURCE DETAILS

All models are trained for 25 epochs using AdamP optimizer (Heo et al., 2021) by setting the initial learning rate as 0.0005 and weight decay as 0.0001. The learning rate is decayed by a factor of 0.1 for the last 10 epochs. Following Chen et al. (2021), different learning rate multipliers are applied for the visual backbone ($\times 0.01$) and the textual backbone ($\times 0.1$). The visual backbone is frozen for 2 epochs, and a linear learning rate warmup is applied for the first epoch after the freezing. Also, layer-wise learning rate decay (LLRD) for each transformer block is applied by 0.7. The batch size is set to 128. Lastly, for the generalizability of GPO, SizeAugment is employed as Chen et al. (2021).

The hyperparameters of PCME++ are set as follows; the affine transform is initialized by $a = b = 5$ in Equation (2); $\alpha$ for pseudo-positives as 0.1; VIB $\beta$ as 0.0001. PCME++ mixes 25% of images in the mini-batch by Mixup or CutMix with a mixing ratio drawn from Beta$(2, 2)$. For comparison methods, The triplet loss margin is set to 0.2 (for VSE$\infty$ (Chen et al., 2021)) and the initial softmax temperature for InfoNCE (Radford et al., 2021) is set to 1.0. PCME (Chun et al., 2021) uses the same initialization of PCME++ for affine transform and VIB, while 8 samples are drawn per input for computing matching probability.

We use different hyperparameters for the large-scale pre-training task in Section 3.5. As our implementation is based on `openclip` (Ilharco et al., 2021), we generally follow the official `openclip` hyperparameters. More details are in Appendix C.6.

Table B.1 shows the detailed hyperparameter settings and the detailed GPU resource information.

Table C.1: **Comparisons with state-of-the-art models.** All numbers are reproduced by the official weights. We highlight **the best scores** except expensive retrieval methods, such as BLIP.

| Method | Efficient retrieval? | ECCV Caption mAP@R | ECCV Caption R-P | ECCV Caption R@1 | CxC R@1 | COCO 1K R@1 | COCO 5K R@1 | RSUM |
|---|---|---|---|---|---|---|---|---|
| CVSE (Wang et al., 2020) | ✔ | 37.4 | 47.5 | 76.7 | 45.8 | 67.0 | 43.8 | 511.1 |
| VSRN (Li et al., 2019) | ✔ | 42.3 | **51.8** | 81.5 | 48.9 | 69.5 | 46.7 | 515.9 |
| NCR (Huang et al., 2021) | ✔ | 36.4 | 46.3 | 79.9 | 51.8 | 71.0 | 50.0 | 522.6 |
| VSE∞ (BUTD region) (Chen et al., 2021) | ✔ | 40.5 | 50.0 | 82.5 | 52.4 | 72.2 | 50.4 | 527.5 |
| VSE∞ (WSL) | ✔ | **42.4** | 51.4 | 86.4 | 60.8 | 78.3 | 59.0 | 545.1 |
| VSE∞ (B/16, our implementation) | ✔ | 41.7 | 50.6 | 86.3 | 62.3 | 79.1 | 60.7 | 547.2 |
| ViLT (Kim et al., 2021) | ✘ | 34.6 | 44.3 | 77.8 | 53.7 | 72.8 | 52.2 | 528.6 |
| VinVL (Zhang et al., 2021c) | ✘ | 40.8 | 49.6 | 87.8 | 67.8 | 82.4 | 66.4 | 555.5 |
| BLIP (Li et al., 2022b) | ✘ | 40.5 | 48.4 | 91.0 | 74.3 | 86.1 | 73.1 | 564.4 |
| CLIP Zero-shot (L/14) (Radford et al., 2021) | ✔ | 28.0 | 37.8 | 72.2 | 48.1 | 64.8 | 46.4 | 491.6 |
| PCME++ (B/16) | ✔ | 42.2 | 51.2 | 86.6 | 62.9 | 79.6 | 61.3 | 548.5 |
| PCME++ (L/14) | ✔ | 42.1 | 50.8 | **88.8** | **65.9** | **81.8** | **64.3** | **554.7** |

### B.3 SWA AND MODEL SELECTION

For the evaluation, the best model based on the validation rsum is selected. However, the clean validation set is not always achievable; *e.g.*, if the dataset has a significant distribution shift. For example, as we know that the COCO Caption is noisy due to a lot of FNs (Chun et al., 2022), validating using the noisy annotations could lead to underperforming models. Instead of the best validation model selection, we can apply SWA (Izmailov et al., 2018), where it does not need an additional optimization process to the existing training procedure, but only training weight trajectory (*i.e.*, weights per every training epoch) is required. SWA is a weight average method of sufficiently trained weights. SWA aims to achieve flat minima, as well as more generalizable and robust solutions (Cha et al., 2021). When we apply SWA to our model, we apply SWA for the last 10 epochs. Although the SWA models are not compared to the other models due to the fair comparison issue, I strongly encourage the use of SWA for future research.

## C ADDITIONAL EXPERIMENTAL RESULTS

### C.1 COMPARISONS WITH STATE-OF-THE-ARTS

Table C.1 shows the comparisons of PCME++ and state-of-the-arts with different backbones. Note that ViLT (Kim et al., 2021), VinVL (Zhang et al., 2021c) and BLIP (Li et al., 2022b) need heavy computations to perform retrieval because they have to compute pair-wise similarity for all pairs. For example, they need $O(5,000 \times 25,000)$ computation budgets for measuring retrieval performances. On the other hand, methods with separated encoders just need $O(5,000 + 25,000)$ computation budgets, 4,166 times smaller computation budgets compared to expensive retrieval methods. Therefore, the table only highlights the best retrieval performances among efficient retrieval methods for a fair comparison. Note that even in an expensive cross-attention architecture, a probabilistic approach can be beneficial, as shown by MAP (Ji et al., 2023). PCME++ is applicable to MAP by replacing the 2-Wasserstein distance with CSD. However, as it is out of scope of this work, the comparison with MAP is omitted in this paper. PCME++ achieves the best recall scores for all evaluation benchmarks while showing second and third-best ECCV mAP@R and R-Precision. One possible explanation is the capability of the backbone architecture. For example, VSE∞ with CLIP B/16 backbone shows much better recall scores than VSE∞ with WSL backbone, but VSE∞ (WSL) shows better mAP@R and R-Precision than the CLIP backbone. From this observation, we expect that PCME++ can outperform the previous retrieval methods in precision metrics if we train PCME++ using different backbones, such as large-scale weakly supervised learning (WSL) backbone (Mahajan et al., 2018).

### C.2 THE EFFECT OF PSEUDO-POSITIVES (PPS)

We evaluate various $\alpha$ under various noise ratios (NR). Table C.2 shows two findings: (1) unlike "no noise ratio" case (*i.e.*, Table C.3), if there exist noisy correspondences (NC), PPs can harm the performances. One possible explanation is that PPs can be incorrect if there are too many NCs. (2) When we tune $\alpha$, we obtain the best performances in the 50% NR scenario. However, we observe that a model is easily overfitted when we weaken the strength of PPs. Figure C.1a shows that when

| Noise ratio | PP $\alpha$ | EC mAP | EC R-P | EC R@1 | COCO 1K R@1 | COCO 5K R@1 | RSUM |
|---|---|---|---|---|---|---|---|
| | 0 | **37.9** | **47.8** | 79.0 | **71.9** | **50.9** | **526.0** |
| 20% | 0.01 | 37.7 | 47.6 | **80.0** | 71.6 | 50.4 | 524.6 |
| | 0.1 | **37.9** | 47.7 | **79.7** | 70.8 | 49.5 | 522.4 |
| | 0 | **35.7** | **45.8** | **76.3** | **67.6** | **45.5** | **511.0** |
| 50% | 0.01 | 35.2 | 45.3 | 75.1 | 66.1 | 43.4 | 506.8 |
| | 0.1 | 34.4 | 44.6 | 75.0 | 65.7 | 44.0 | 503.9 |

Table C.2: **different noisy ratio results with varying PP $\alpha$. $\alpha = 0.1$ equals "PCME++ (ours)" in Table 2.**

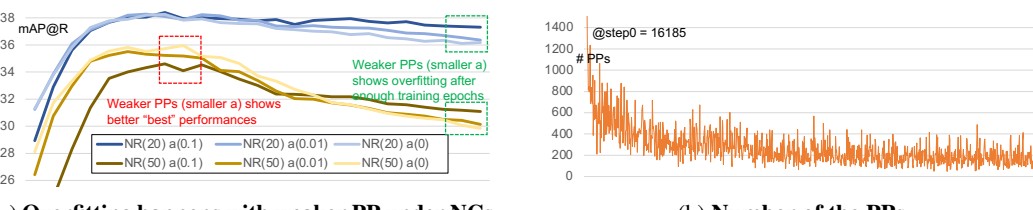

(a) **Overfitting happens with weaker PP under NCs.**  (b) **Number of the PPs.**

Figure C.1: **The effect of Pseudo-Positives (PPs) during training.**

$\alpha = 0$, the best performed score is better than $\alpha = 0.1$, but at the end of the training, $\alpha = 0.1$ shows a weaker overfitting. As the current evaluation criterion is based on the validation-based model selection, even though PPs can prevent overfitting, PPs cannot be directly helpful for achieving the best performances under NCs. In other words, PPs are helpful for preventing overfitting and gradient vanishing (Section 2.3); thereby, when there is less noise (*i.e.*, no noise ratio scenario), PPs can improve performances with a gap, especially for handling false negatives (FNs) well (See mAP@R scores of 2nd and 4th row in Table 3 and Table C.3).

The number of PPs during training is illustrated in Figure C.1b. We observe that # PPs is 16,185 at the first iteration, but it has converged to around 260 after 1 epoch.

## C.3   MORE ABLATION STUDIES

Table C.3: **Pseudo-positive $\alpha$ ablation study.**

| | ECCV Caption | | | CxC | COCO | | |
|---|---|---|---|---|---|---|---|
| $\alpha$ | mAP@R | R-P | R@1 | R@1 | 1K R@1 | 5K R@1 | RSUM |
| 0.1 | 40.2 | 49.8 | 83.1 | 56.5 | 75.1 | 54.8 | 536.0 |
| 0.5 | 40.0 | 49.5 | 83.1 | 56.7 | 75.4 | 55.0 | 536.8 |
| 2 | 40.1 | 49.7 | 83.0 | 56.5 | 75.1 | 54.8 | 535.8 |
| 5 | 40.3 | 49.9 | 83.1 | 55.7 | 74.7 | 53.9 | 534.9 |
| 10 | 40.2 | 49.9 | 82.5 | 54.5 | 73.7 | 52.6 | 531.9 |

**PP and MSDA.**   Table C.3 shows the ablation study for pseudo positive $\alpha$. The table shows that PCME++ is not very sensitive to the choice of $\alpha$. We choose $\alpha = 0.1$, which shows the second-best EC mAP@R and COCO recall measures. Table C.4 shows the ablation study for the mixed sample data augmentation design choice. The design choice for PCME++ shows the best performance.

**VIB.**   The parameter study of VIB $\beta$ is provided in Table C.5. In the table, the average uncertainty quantification $\|\sigma\|_1$, the retrieval performances, and the correlation coefficient between the uncertainty and the average R@1, $\rho$, by varying VIB $\beta$ from 0 to 0.001 (where $\times 1 = 0.0001$) are reported. $\rho = -1$ means that uncertainty and R@1 are perfectly negatively correlated, *i.e.*, a higher uncertainty shows a lower recall (what we expect). If $\rho = 0$, then the uncertainty quantity and the recall performance are not correlated. Note that although we chose RSUM for performance comparisons due to the space limitation, we can observe a similar phenomenon for any other metric.

Table C.4: **MSDA ablation study.**

| Mixup $\lambda$ | CutMix $\lambda$ | Mix ratio | in-batch? | ECCV Caption mAP@R | R-P | R@1 | CxC R@1 | COCO 1K R@1 | 5K R@1 | RSUM |
|---|---|---|---|---|---|---|---|---|---|---|
| 2 | 0 | 25% | ✔ | 37.3 | 47.3 | 79.6 | 51.9 | 71.7 | 50.0 | 525.5 |
| 0 | 2 | 25% | ✔ | 37.5 | 47.6 | 79.2 | 50.5 | 70.9 | 48.8 | 523.4 |
| 1 | 1 | 50% | ✘ | 39.7 | 49.5 | 81.8 | 55.2 | 74.5 | 53.5 | 534.0 |
| 1 | 1 | 25% | ✘ | 39.9 | 49.6 | 82.3 | 55.5 | 74.6 | 53.8 | 534.4 |
| 2 | 2 | 25% | ✘ | 40.0 | 49.6 | 82.8 | 55.8 | 74.5 | 54.0 | 534.4 |
| 1 | 1 | 50% | ✔ | 39.9 | 49.6 | 82.7 | 55.4 | 74.4 | 53.7 | 534.1 |
| 2 | 2 | 25% | ✔ | **40.1** | **49.7** | **82.9** | **56.5** | **75.0** | **54.7** | **535.9** |

Table C.5: **VIB $\beta$ ablation study.** $\times 1$ denotes the paper's choice (0.0001).

| VIB $\beta$ | ECCV Caption mAP@R | R-P | R@1 | CxC R@1 | COCO 1K R@1 | 5K R@1 | RSUM | $\|\sigma\|_1$ | $-\rho$ |
|---|---|---|---|---|---|---|---|---|---|
| $\times 0$ | 39.3 | 49.0 | 83.1 | 56.1 | 74.5 | 54.3 | 534.5 | $2\times 10^{-4}$ | 0.76 |
| $\times 0.1$ | 39.8 | 49.4 | 83.2 | 57.1 | 75.6 | 55.4 | 537.4 | 1.1 | 0.87 |
| $\times 0.2$ | 39.7 | 49.2 | 83.6 | 57.1 | 75.6 | 55.3 | 537.4 | 2.2 | 0.91 |
| $\times 0.5$ | 39.7 | 49.3 | 82.9 | 56.7 | 75.5 | 55.0 | 537.2 | 4.2 | 0.92 |
| $\times 1$ | **40.1** | **49.7** | **83.1** | **56.8** | **75.4** | **55.1** | **537.0** | **7.1** | **0.94** |
| $\times 2.0$ | 40.0 | 49.6 | 82.6 | 56.7 | 75.4 | 55.0 | 536.7 | 11.6 | 0.95 |
| $\times 5.0$ | 40.1 | 49.7 | 83.2 | 56.1 | 74.8 | 54.3 | 535.5 | 23.1 | 0.91 |
| $\times 10.0$ | 40.1 | 49.7 | 83.0 | 55.4 | 74.6 | 53.6 | 534.4 | 37.6 | 0.9 |

Table C.6: **Impact of architecture design choice.** Details are the same as the previous tables.

| # layers for $\log \sigma^2$ | GPO | ECCV Caption mAP@R | R-P | R@1 | CxC R@1 | COCO 1K R@1 | 5K R@1 | RSUM |
|---|---|---|---|---|---|---|---|---|
| 1 | ✘ | 37.4 | 47.4 | 79.2 | 51.0 | 70.4 | 49.2 | 521.8 |
| 2 | ✔ | **40.2** | **49.7** | 83.2 | 56.6 | 75.3 | 54.8 | 536.5 |
| 1 | ✔ | 40.0 | 49.6 | **83.3** | **57.0** | **75.5** | **55.3** | **537.1** |

Table C.5 shows three observations. (1) In terms of performance, there exists a sweet spot between $\beta$ $\times 0.1$ and $\times 2$, but the performance drop is relatively not significant (cf. RSUM of VSE$\infty$ and PCME are 536.5 and 532.0, respectively). (2) The average uncertainty quantity $\|\sigma\|_1$ is increased by the $\beta$ value. It supports that VIB regularizes probabilistic embeddings from variance collapse ($\sigma \to 0$). (3) if we did not use VIB ($\beta = 0$), the correlation between uncertainty and recall is the smallest (-0.76), while a proper choice of $\beta$ improves $\rho$, *e.g.*, $-0.92$ ($\times 0.2$, $\times 0.5$), $-0.95$ ($\times 1$, $\times 2$). Our choice $\times 1$ (*i.e.*, 0.0001) shows the reasonable RSUM score (537.0, while the best is 538.2) and the second best $\rho$ (-0.94, while the best is -0.95).

**Architecture.** Table C.6 shows the architecture ablation study: (1) GPO improves overall performances; (2) if we use a more complex $\log \sigma^2$ head, ECCV Caption metrics are slightly improved by capturing ambiguity caused by FNs well. However, the performance improvements are marginal, and it shows inferior R@$k$ scores than a shallower $\log \sigma^2$ head. Therefore, PCME++ uses the number of layers for the $\log \sigma^2$ head as 1.

## C.4 T-SNE VISUALIZATION DETAILS

For Figure 4, t-SNE (Van der Maaten & Hinton, 2008) is respectively applied for PCME++ and VSE$\infty$ embeddings extracted from the COCO Caption test split. Then, three images and their corresponding captions (the same colored image and caption are a "positive" pair; otherwise, a pair is negative) in each embedding space are illustrated in the figure.

The purpose of Figure 4 is to show how the learned embedding space by PCME++ can successfully capture the inherent ambiguity in the dataset. Qualitatively, PCME++ embedding space captures the

Table C.7: **Effect of inference methods.** We compare the mean-only inference and probability distance-based inferences using our ViT-B/32 SWA model. Each number is the average of different three runs.

| Method | Prob? | ECCV Caption | | | CxC | | COCO | |
| | | mAP@R | R-P | R@1 | R@1 | 1K R@1 | 5K R@1 | RSUM |
|---|---|---|---|---|---|---|---|---|
| Mean only | ✗ | **40.2** | 49.8 | 83.5 | 56.9 | 75.2 | 55.2 | 536.3 |
| 2-Wasserstein | ✔ | **40.2** | **49.9** | 83.0 | 56.6 | 75.2 | 54.8 | 535.9 |
| CSD (ours) | ✔ | **40.2** | 49.8 | **83.6** | **57.2** | **75.6** | **55.5** | **537.3** |
| FAISS (Meany only) | ✗ | 40.1 | 49.7 | 83.5 | 56.4 | 74.7 | 54.6 | 531.2 |
| FAISS + $\sigma$ re-ranking | ✔ | 40.1 | 49.7 | 83.2 | **56.6** | **74.8** | **54.8** | **531.7** |

ambiguity of the many-to-many relationships despite the false negatives. However, the figure can be misleading whether the purpose of PCME++ is to make "overlap" between two distributions.

The overlap between the two distributions itself is not directly related to the uncertainty. Conceptually, the overlap between two Gaussian distributions can be represented as the Bhattacharyya coefficient (or Bhattacharyya distance). Here, we recall the CSD's main property (b) discussed in Section 2.2: if the match $m_{vt}$ is certain, then $Z_v$ and $Z_t$ have small variances. As discussed in Section 2.2, a similarity function, that measures whether two distributions are the same or not, cannot satisfy the property because it cannot handle the case when the distributions have the same $\mu$, but larger $\sigma$. There is no motivation to reduce the size of $\sigma$ using the distance. As shown in Table 4, Bhattacharyya distance is not an effective probabilistic distance for learning probabilistic embeddings as much as CSD. On the other hand, the learned embedding space by CSD-trained PCME++ is a reasonable probabilistic space. Table C.7 shows that even though we use Wasserstein distance as the similarity function of retrieval, the overall precision-based retrieval performances are almost preserved; it means that the probabilistic space learned by PCME++ is a sound metric space of Wasserstein distance.

Finally, instead of focusing on the overlap between two distributions, we focus on how CSD can learn the embedding space shown in Figure 4. We recall the properties of our desired probabilistic embedding space: (1) if the correspondence between a given image and caption is certain (*i.e.*, they are certainly positive or negative), then the variance of each instance should be small, (2) if the correspondence is uncertain (*i.e.*, the match is sometimes positive and sometimes negative. It can be happened due to the false negatives in the dataset as shown in Figure 1 and Section 2.1), then the variance of each instance should be large. As mentioned before, CSD can give proper supervision for the desired embedding space. For example, let's approximate the plane figures in Figure 4 to a visual embedding $v$ and the plane captions to a textual embedding $t$ (because they have very high semantic similarity). In this case, by choosing different image-caption pairs, the supervision between $v$ and $t$ can be either positive or negative because our training dataset only has one-to-one positive correspondences. In this case, our objective function enforces the CSD between the matches to be larger until the penalty for the positive supervision and the negative supervision are balanced. As shown in Figure 4, the visual embeddings and textual embeddings are aligned into almost the same place by making the $\mu$ distance closer but penalizing the uncertain supervision by enlarging $\sigma$.

## C.5 COMPARISONS OF DIFFERENT RETRIEVAL STRATEGIES

A modified ANN for PCME++ is a two-step process. First, an Euclidean distance-based index system for $\mu$ is built as usual, while $\sigma^2$ is saved into key-value storage. Then, $K$ items are retrieved by performing ANN on the $\mu$ index. Lastly, the retrieved items are re-ranked by computing the summation of the $\mu$ distance and $\sigma^2$ value of the retrieved items.

Table C.7 shows the comparisons of different retrieval strategies using PCME++ B/32 model. "Mean only" denotes the retrieval strategy only using $\mu$ vectors, without $\sigma$. "2-Wasserstein" and "CSD" denote that each probabilistic distance is used for the retrieval. In the table, we observe that mean-only retrieval shows sufficiently good performances, but using CSD improves the overall performance.

This paper additionally shows the approximated KNN (ANN) results using FAISS. First, a FAISS search index using $\mu$ vectors is built. Then, ANN is performed on the FAISS index to get the ranked list. Finally, the ranked list is re-ranked by CSD. Here, CSD can be efficiently computed by storing

gallery $\sigma$ into a fast key-value storage, such as Redis. As shown in the table, ANN can be efficiently and effectively applied to PCME++ with a reasonable computation-performance trade-off.

## C.6 Details of automatic prompt-filtering by PCME++

For the experiments, a randomly initialized ViT-B/16 is trained by InfoNCE loss and PCME++ loss on Conceptual Caption 3M (Sharma et al., 2018), 12M (Changpinyo et al., 2021) and RedCaps (Desai et al., 2021) using hyperparameters in Table B.1. The implementation is based on openclip (Ilharco et al., 2021) software. For a fair comparison, the PCME++ model has the same architecture as the vanilla CLIP model except for the additional uncertainty head (as described in Figure 2). For the stable training, the original CLIP loss and the PCME++ loss are used at the same time; solely using PCME++ loss also converges but shows much worse ZS performance. Note that we cannot apply PPs and MSDA for this experiment due to the additional CLIP loss. We also set longer warmup steps than the original setting ($\times 5$ for PCME++). The pre-trained models are evaluated on the ImageNet (Russakovsky et al., 2015) zero-shot (ZS) classification task. Specifically, 80 prompts provided by CLIP (Radford et al., 2021) (shown in the below) are used for the ZS classification. In Table 5, "A photo of a $\{\cdot\}$" denotes that only "A photo of a $\{\cdot\}$" prompt is used for the zero-shot classification, while "All 80 prompts" denotes that all 80 prompts are used for computing text embeddings and the average text embedding is used for the zero-shot classification.

**80 base prompts.** `a photo of a {}., a bad photo of a {}., a photo of many {}., a sculpture of a {}., a photo of the hard to see {}., a low resolution photo of the {}., a rendering of a {}., graffiti of a {}., a bad photo of the {}., a cropped photo of the {}., a tattoo of a {}., the embroidered {}., a photo of a hard to see {}., a bright photo of a {}., a photo of a clean {}., a photo of a dirty {}., a dark photo of the {}., a drawing of a {}., a photo of my {}., the plastic {}., a photo of the cool {}., a close-up photo of a {}., a black and white photo of the {}., a painting of the {}., a painting of a {}., a pixelated photo of the {}., a sculpture of the {}., a bright photo of the {}., a cropped photo of a {}., a plastic {}., a photo of the dirty {}., a jpeg corrupted photo of a {}., a blurry photo of the {}., a photo of the {}., a good photo of the {}., a rendering of the {}., a {} in a video game., a photo of one {}., a doodle of a {}., a close-up photo of the {}., the origami {}., the {} in a video game., a sketch of a {}., a doodle of the {}., a origami {}., a low resolution photo of a {}., the toy {}., a rendition of the {}., a photo of the clean {}., a photo of a large {}., a rendition of a {}., a photo of a nice {}., a photo of a weird {}., a blurry photo of a {}., a cartoon {}., art of a {}., a sketch of the {}., a embroidered {}., a pixelated photo of a {}., itap of the {}., a jpeg corrupted photo of the {}., a good photo of a {}., a plushie {}., a photo of the nice {}., a photo of the small {}., a photo of the weird {}., the cartoon {}., art of the {}., a drawing of the {}., a photo of the large {}., a black and white photo of a {}., the plushie {}., a dark photo of a {}., itap of a {}., graffiti of the {}., a toy {}., itap of my {}., a photo of a cool {}., a photo of a small {}., a tattoo of the {}.`

This paper explores the potential of PCME++ for automatic prompt-filtering with simple uncertainty-based filtering. First, the prompts for every class are sorted by their uncertainty, *i.e.*, $\|\sigma\|_1$. Then, uncertain prompts are filtered out, and the remaining prompts are used for ZS classification. Here, two strategies are tested. First, the same top-K uncertain prompts for all classes are filtered. As shown in Figure C.2a, this strategy slightly improves the overall performances, but it only shows a marginal improvement against the "all" baseline (+0.04%). To further improve the uncertainty-based filtering, the strategy with different top-K for different prompts is also explored. As shown in Table 5, this strategy shows very effective performance improvement against the baseline (+5.42%). Figure C.2b shows the detailed population of the best top-K filtering per class. Here, the classes whose accuracy is 0% are omitted. Interestingly, we observe that 10% of classes (105) show the best ZS performances when all 80 prompts are used. On the other hand, about half of the classes (499) show the best performance when more than 35 prompts are filtered out.

Figure C.4 shows the examples of the filtered prompts and the ImageNet validation images. Interestingly, we can observe that the keywords `"cropped"` or `"close-up"` are selected for the Jack-o'lantern class due to the low-quality images of the class. On the other hand, a generic class, such as rapeseed, shows various prompts to cover its visual diversity.

This primitive study on uncertainty-based prompt-filtering has a limitation. This study has no validation split, *i.e.*, the best top-K prompt for each class is directly searched from the validation

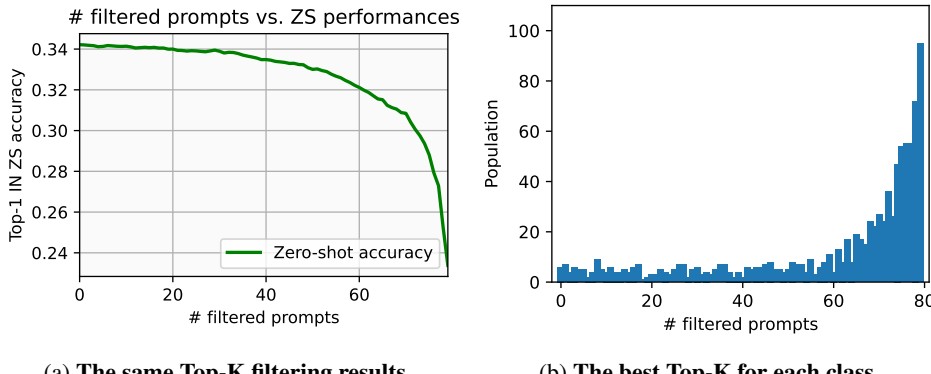

(a) **The same Top-K filtering results.**  (b) **The best Top-K for each class.**

Figure C.2: **Automatic prompt-filtering results.** (a) shows the ImageNet (IN) zero-shot (ZS) results when prompts are filtered by the same top-K for every class. Applying all the same top-K filtering does not improve ZS performances. (b) shows the population of best top-K filtering for all classes. Here, 906 of classes among 1,000 classes show the best performance when using less than 10 prompts.

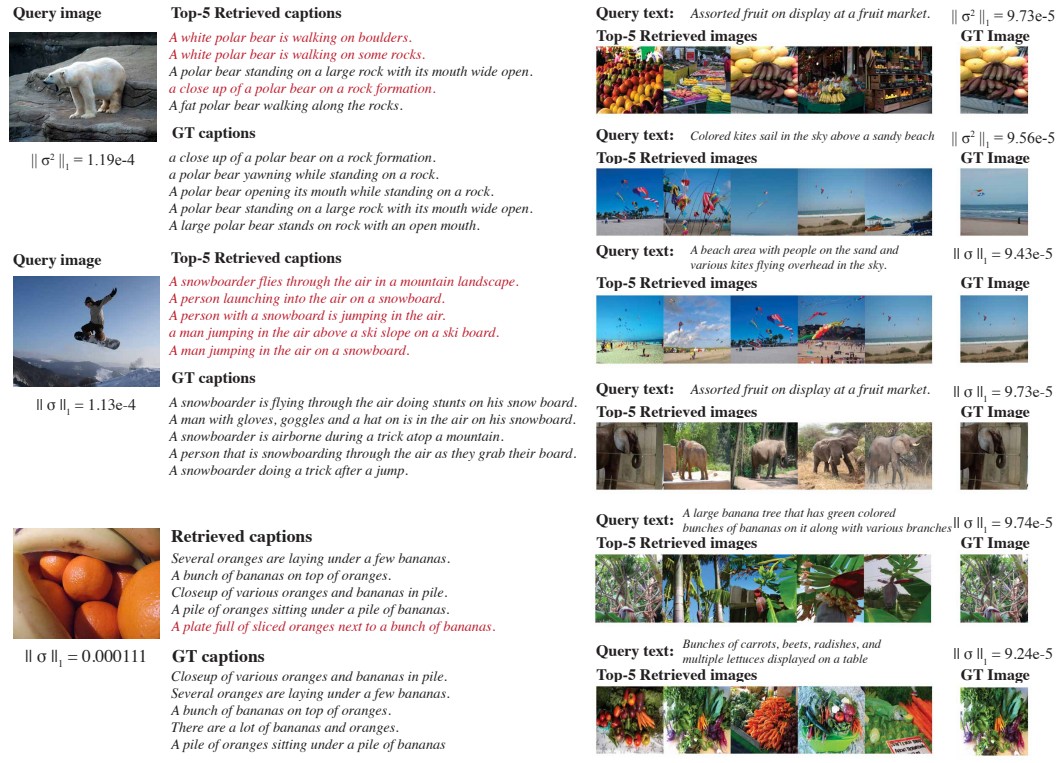

Figure C.3: **Example of images and captions with high uncertainty.**

split. Searching for the best top-K for each class without directly tuning on test split using strong probabilistic pre-trained image-text representations will be an interesting future research direction.

## C.7 FULL EXPERIMENTAL RESULTS

The image-to-text and text-to-image R@5 and R@10 results are shown in Table C.8 and Table C.9. The full experimental results, including separated image-to-text and text-to-image retrieval results for the main table and standard errors, are included in Table C.10 and Table C.11. The full experimental numbers for all experiments can be found in https://naver-ai.github.io/pcmepp/.

Table C.8: **Image-to-text retrieval R@5 and R@10 results.**

| Backbone | Method | CxC | | | | COCO | |
|---|---|---|---|---|---|---|---|
| | | R@5 | R@10 | 1K R@5 | 1K R@10 | 5K R@5 | 5K R@10 |
| ViT-B/32 (151M) | VSE∞ | 87.1 | 93.4 | 96.7 | 98.8 | 85.4 | 92.2 |
| | P2RM | 85.5 | 92.2 | 96.1 | 98.6 | 83.5 | 90.9 |
| | DAA | 86.75 | 93.25 | 96.45 | 98.85 | 84.95 | 92.0 |
| | InfoNCE | 87.3 | 93.4 | 96.5 | 98.9 | 85.9 | 92.3 |
| | PCME | 87.5 | 93.5 | 96.6 | 98.7 | 85.8 | 92.3 |
| | PCME++ ($\mu$ only) | 88.5 | 94.2 | 97.0 | 99.0 | 87.1 | 93.2 |
| | PCME++ | 88.4 | 94.0 | 97.0 | 99.0 | 87.0 | 93.0 |
| | PCME++ (SWA) | 88.5 | 94.0 | 97.0 | 99.0 | 87.1 | 92.9 |
| ViT-B/16 (150M) | VSE∞ | 91.1 | 95.6 | 97.8 | 99.4 | 89.9 | 94.8 |
| | P2RM | 86.0 | 92.7 | 96.2 | 98.7 | 84.3 | 91.5 |
| | DAA | 53.9 | 67.1 | 76.8 | 87.6 | 49.9 | 62.7 |
| | InfoNCE | 90.9 | 95.8 | 97.8 | 99.3 | 89.7 | 94.9 |
| | PCME | 90.5 | 95.4 | 97.7 | 99.3 | 89.2 | 94.5 |
| | PCME++ ($\mu$ only) | 91.5 | 95.8 | 97.9 | 99.3 | 90.3 | 95.1 |
| | PCME++ | 91.3 | 95.7 | 97.9 | 99.3 | 90.1 | 95.0 |
| | PCME++ (SWA) | 91.5 | 95.9 | 97.9 | 99.3 | 90.4 | 95.1 |
| ViT-L/14 (428M) | VSE∞ | 58.8 | 72.6 | 82.2 | 91.4 | 55.7 | 69.4 |
| | InfoNCE | 82.8 | 91.7 | 95.3 | 98.6 | 80.2 | 90.0 |
| | PCME | 91.8 | 95.9 | 98.1 | 99.4 | 90.7 | 95.2 |
| | PCME++ | 93.4 | 96.8 | 98.5 | 99.6 | 92.2 | 96.2 |

Table C.9: **Text-to-image retrieval R@5 and R@10 results.**

| Backbone | Method | CxC | | | | COCO | |
|---|---|---|---|---|---|---|---|
| | | R@5 | R@10 | 1K R@5 | 1K R@10 | 5K R@5 | 5K R@10 |
| ViT-B/32 (151M) | VSE∞ | 77.7 | 86.5 | 92.2 | 96.7 | 75.5 | 84.8 |
| | P2RM | 77.2 | 86.2 | 92.3 | 96.8 | 74.9 | 84.4 |
| | DAA | 76.9 | 86.1 | 91.9 | 96.5 | 74.7 | 84.2 |
| | InfoNCE | 77.3 | 86.5 | 92.3 | 96.9 | 75.1 | 84.7 |
| | PCME | 77.3 | 86.4 | 92.1 | 96.9 | 75.0 | 84.6 |
| | PCME++ | 78.5 | 87.1 | 92.8 | 97.1 | 76.5 | 85.4 |
| | PCME++ (SWA) | 78.6 | 87.3 | 92.8 | 97.1 | 76.5 | 85.5 |
| ViT-B/16 (150M) | VSE∞ | 82.0 | 89.5 | 94.2 | 97.5 | 80.3 | 88.2 |
| | P2RM | 78.7 | 87.3 | 93.0 | 97.2 | 76.6 | 85.7 |
| | DAA | 50.3 | 62.5 | 73.4 | 85.1 | 47.1 | 59.1 |
| | InfoNCE | 81.3 | 89.1 | 94.0 | 97.7 | 79.5 | 87.7 |
| | PCME | 80.9 | 88.9 | 93.9 | 97.7 | 79.1 | 87.5 |
| | PCME++ | 82.0 | 89.7 | 94.4 | 97.8 | 80.3 | 88.3 |
| | PCME++ (SWA) | 82.1 | 89.7 | 94.4 | 97.8 | 80.4 | 88.4 |
| ViT-L/14 (428M) | VSE∞ | 46.4 | 61.1 | 74.2 | 87.4 | 42.9 | 57.1 |
| | InfoNCE | 73.6 | 84.2 | 91.3 | 96.4 | 71.0 | 82.3 |
| | PCME | 82.7 | 90.2 | 94.5 | 97.8 | 81.1 | 88.8 |
| | PCME++ | 84.0 | 90.8 | 95.1 | 98.1 | 82.6 | 89.7 |

Table C.10: **Image-to-text retrieval full results.** P2RM ViT-B/32 result has no standard error because we failed to train multiple P2RM due to its instability. Full numbers also can be found in https://github.com/naver-ai/pcmepp.

| Backbone | Method | ECCV Caption | | | CxC | COCO | |
|---|---|---|---|---|---|---|---|
| | | mAP@R | R-P | R@1 | R@1 | 1K R@1 | 5K R@1 |
| ViT-B/32 (151M) | VSE$\infty$ | 32.3 $(\pm 0.2)$ | 43.3 $(\pm 0.1)$ | 77.2 $(\pm 0.4)$ | 63.8 $(\pm 0.4)$ | 82.0 $(\pm 0.3)$ | 62.3 $(\pm 0.5)$ |
| | P2RM | 30.2 $(\cdot)$ | 41.7 $(\cdot)$ | 72.2 $(\cdot)$ | 58.1 $(\cdot)$ | 78.6 $(\cdot)$ | 56.6 $(\cdot)$ |
| | DAA | 31.2 $(\pm 0.1)$ | 42.3 $(\pm 0.1)$ | 75.9 $(\pm 0.3)$ | 61.5 $(\pm 0.3)$ | 79.8 $(\pm 0.3)$ | 59.8 $(\pm 0.1)$ |
| | InfoNCE | 31.2 $(\pm 0.1)$ | 42.3 $(\pm 0.1)$ | 75.4 $(\pm 1.1)$ | 61.8 $(\pm 0.1)$ | 80.3 $(\pm 0.6)$ | 60.1 $(\pm 0.2)$ |
| | PCME | 31.2 $(\pm 0.0)$ | 42.3 $(\pm 0.0)$ | 74.9 $(\pm 0.3)$ | 61.5 $(\pm 0.6)$ | 80.1 $(\pm 0.2)$ | 59.9 $(\pm 0.6)$ |
| | PCME++ ($\mu$ only) | 32.1 $(\pm 0.2)$ | 43.1 $(\pm 0.2)$ | 77.4 $(\pm 1.4)$ | 64.1 $(\pm 0.5)$ | 82.0 $(\pm 0.2)$ | 62.5 $(\pm 0.4)$ |
| | PCME++ | 32.3 $(\pm 0.2)$ | 43.4 $(\pm 0.3)$ | 76.6 $(\pm 0.6)$ | 63.5 $(\pm 0.5)$ | 81.6 $(\pm 0.4)$ | 62.1 $(\pm 0.6)$ |
| | PCME++ (SWA) | 32.5 $(\pm 0.2)$ | 43.6 $(\pm 0.2)$ | 76.3 $(\pm 0.3)$ | 63.4 $(\pm 0.3)$ | 81.8 $(\pm 0.6)$ | 62.0 $(\pm 0.5)$ |
| ViT-B/16 (150M) | VSE$\infty$ | 34.4 $(\pm 0.1)$ | 44.8 $(\pm 0.2)$ | 81.2 $(\pm 0.7)$ | 69.4 $(\pm 0.2)$ | 84.9 $(\pm 0.4)$ | 68.0 $(\pm 0.1)$ |
| | P2RM | 30.6 $(\pm 0.2)$ | 42.2 $(\pm 0.0)$ | 72.9 $(\pm 2.2)$ | 58.5 $(\pm 0.4)$ | 78.3 $(\pm 0.4)$ | 56.8 $(\pm 0.3)$ |
| | DAA | 12.4 $(\pm 0.1)$ | 22.4 $(\pm 0.2)$ | 40.3 $(\pm 0.2)$ | 26.4 $(\pm 0.4)$ | 46.1 $(\pm 0.7)$ | 24.3 $(\pm 0.3)$ |
| | InfoNCE | 33.7 $(\pm 0.1)$ | 44.4 $(\pm 0.1)$ | 79.7 $(\pm 0.4)$ | 68.2 $(\pm 0.6)$ | 84.3 $(\pm 0.7)$ | 66.8 $(\pm 0.5)$ |
| | PCME | 33.2 $(\pm 0.3)$ | 44.0 $(\pm 0.4)$ | 79.1 $(\pm 0.4)$ | 66.8 $(\pm 0.6)$ | 83.6 $(\pm 0.3)$ | 65.3 $(\pm 0.6)$ |
| | PCME++ ($\mu$ only) | 34.0 $(\pm 0.1)$ | 44.5 $(\pm 0.3)$ | 80.9 $(\pm 0.6)$ | 69.6 $(\pm 0.8)$ | 85.3 $(\pm 0.2)$ | 68.4 $(\pm 0.7)$ |
| | PCME++ | 34.5 $(\pm 0.1)$ | 45.1 $(\pm 0.1)$ | 81.6 $(\pm 0.2)$ | 69.9 $(\pm 0.2)$ | 85.3 $(\pm 0.1)$ | 68.7 $(\pm 0.4)$ |
| | PCME++ (SWA) | 34.6 $(\pm 0.1)$ | 45.2 $(\pm 0.1)$ | 81.8 $(\pm 0.8)$ | 70.3 $(\pm 0.1)$ | 85.6 $(\pm 0.1)$ | 69.0 $(\pm 0.1)$ |
| ViT-L/14 (428M) | VSE$\infty$ | 15.7 | 27.2 | 39.7 | 28.9 | 51.2 | 27.4 |
| | InfoNCE L/14 | 27.8 | 39.6 | 69.0 | 53.9 | 75.9 | 51.9 |
| | PCME | 34.1 | 44.5 | 81.5 | 70.7 | 86.5 | 69.5 |
| | PCME++ | 35.4 | 45.3 | 84.0 | 73.3 | 87.9 | 71.8 |

Table C.11: **Text-to-image retrieval full results.** Full numbers also can be found in https://github.com/naver-ai/pcmepp.

| Backbone | Method | ECCV Caption | | | CxC | COCO | |
|---|---|---|---|---|---|---|---|
| | | mAP@R | R-P | R@1 | R@1 | 1K R@1 | 5K R@1 |
| ViT-B/32 (151M) | VSE$\infty$ | 47.7 $(\pm 0.2)$ | 55.9 $(\pm 0.3)$ | 88.6 $(\pm 0.9)$ | 49.0 $(\pm 2.6)$ | 67.9 $(\pm 2.2)$ | 46.9 $(\pm 2.6)$ |
| | P2RM | 47.6 $(\cdot)$ | 55.5 $(\cdot)$ | 89.5 $(\cdot)$ | 48.4 $(\cdot)$ | 67.5 $(\cdot)$ | 46.3 $(\cdot)$ |
| | DAA | 47.3 $(\pm 0.4)$ | 55.7 $(\pm 0.4)$ | 88.1 $(\pm 0.3)$ | 48.2 $(\pm 0.1)$ | 67.4 $(\pm 0.1)$ | 46.1 $(\pm 0.1)$ |
| | InfoNCE | 46.8 $(\pm 0.5)$ | 55.1 $(\pm 0.5)$ | 88.0 $(\pm 0.8)$ | 48.0 $(\pm 0.3)$ | 67.7 $(\pm 0.2)$ | 46.0 $(\pm 0.3)$ |
| | PCME | 47.1 $(\pm 0.2)$ | 55.5 $(\pm 0.2)$ | 88.0 $(\pm 0.5)$ | 48.0 $(\pm 0.1)$ | 67.6 $(\pm 0.1)$ | 46.1 $(\pm 0.1)$ |
| | PCME++ ($\mu$ only) | 46.8 $(\pm 0.3)$ | 55.0 $(\pm 0.3)$ | 88.0 $(\pm 0.8)$ | 50.4 $(\pm 0.3)$ | 69.3 $(\pm 0.1)$ | 48.4 $(\pm 0.3)$ |
| | PCME++ | 47.8 $(\pm 0.2)$ | 55.9 $(\pm 0.1)$ | 89.5 $(\pm 0.2)$ | 50.1 $(\pm 0.1)$ | 69.2 $(\pm 0.1)$ | 48.1 $(\pm 0.1)$ |
| | PCME++ (SWA) | 47.8 $(\pm 0.2)$ | 56.0 $(\pm 0.2)$ | 89.5 $(\pm 0.2)$ | 50.2 $(\pm 0.1)$ | 69.3 $(\pm 0.0)$ | 48.3 $(\pm 0.1)$ |
| ViT-B/16 (150M) | VSE$\infty$ | 49.1 $(\pm 0.3)$ | 56.5 $(\pm 0.2)$ | 91.3 $(\pm 0.4)$ | 55.3 $(\pm 0.3)$ | 73.3 $(\pm 0.3)$ | 53.4 $(\pm 0.3)$ |
| | P2RM | 48.8 $(\pm 0.3)$ | 56.8 $(\pm 0.4)$ | 88.5 $(\pm 0.2)$ | 50.0 $(\pm 0.1)$ | 69.2 $(\pm 0.1)$ | 48.1 $(\pm 0.1)$ |
| | DAA | 29.0 $(\pm 0.2)$ | 38.9 $(\pm 0.3)$ | 60.0 $(\pm 1.0)$ | 24.3 $(\pm 0.4)$ | 41.3 $(\pm 0.4)$ | 22.4 $(\pm 0.4)$ |
| | InfoNCE | 48.5 $(\pm 0.2)$ | 56.3 $(\pm 0.1)$ | 89.9 $(\pm 0.2)$ | 53.6 $(\pm 0.3)$ | 72.3 $(\pm 0.1)$ | 51.7 $(\pm 0.3)$ |
| | PCME | 48.7 $(\pm 0.2)$ | 56.5 $(\pm 0.2)$ | 89.5 $(\pm 0.1)$ | 53.1 $(\pm 0.9)$ | 72.0 $(\pm 0.6)$ | 51.2 $(\pm 0.9)$ |
| | PCME++ ($\mu$ only) | 48.5 $(\pm 0.1)$ | 56.3 $(\pm 0.1)$ | 90.4 $(\pm 0.7)$ | 55.4 $(\pm 0.3)$ | 73.4 $(\pm 0.2)$ | 53.6 $(\pm 0.3)$ |
| | PCME++ | 49.7 $(\pm 0.2)$ | 57.2 $(\pm 0.2)$ | 91.4 $(\pm 0.6)$ | 55.2 $(\pm 0.2)$ | 73.4 $(\pm 0.1)$ | 53.4 $(\pm 0.2)$ |
| | PCME++ (SWA) | 49.8 $(\pm 0.1)$ | 57.2 $(\pm 0.2)$ | 91.4 $(\pm 0.7)$ | 55.5 $(\pm 0.2)$ | 73.5 $(\pm 0.1)$ | 53.6 $(\pm 0.2)$ |
| ViT-L/14 (428M) | VSE$\infty$ | 24.7 | 35.8 | 52.7 | 19.7 | 37.9 | 18.0 |
| | InfoNCE L/14 | 43.4 | 52.1 | 82.1 | 42.1 | 63.1 | 39.9 |
| | PCME | 48.2 | 56.0 | 90.5 | 56.1 | 74.1 | 54.3 |
| | PCME++ | 48.6 | 56.3 | 92.5 | 58.9 | 75.8 | 57.1 |

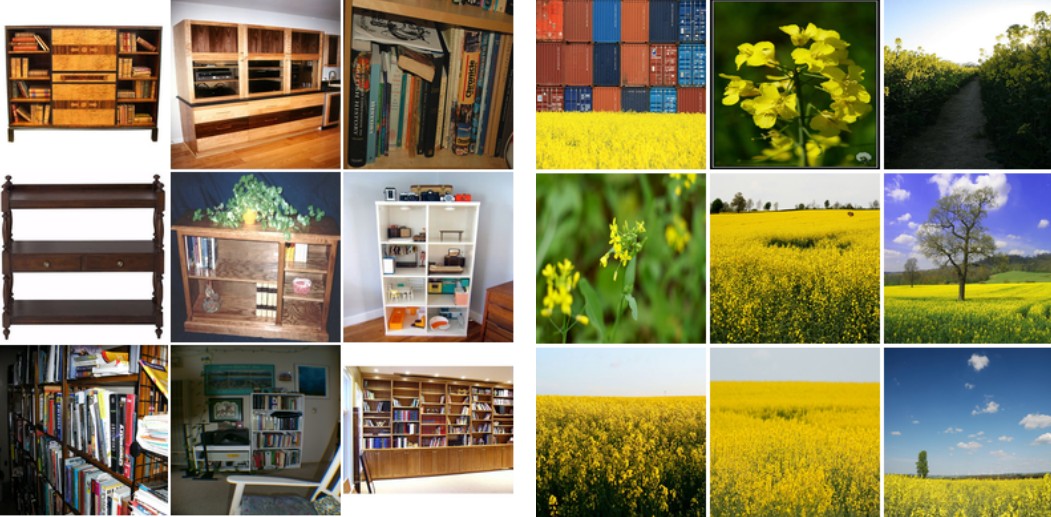

**Label: Bookcase**, selected prompts: `a close-up photo of a {}, a close-up photo of the {}`

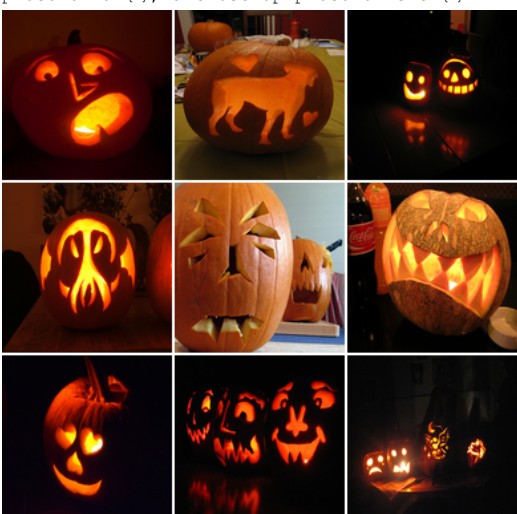

**Label: Jack-o'-lantern**, selected prompts: `a photo of the clean {}. art of the {}. a cropped photo of the {}. a close-up photo of the {}. a photo of a clean {}. a cropped photo of a {}`

**Label: Rapeseed**, selected prompts: `a sculpture of the {}. a {} in a video game. a sculpture of a {}. art of the {}. the {} in a video game. a tattoo of the {}. graffiti of the {}. the plushie {}. a tattoo of a {}. a drawing of a {}. a drawing of the {}. a sketch of the {}. a close-up photo of the {}. art of a {}. a photo of a clean {}. a plushie {}. a close-up photo of a {}. a photo of the clean {}. a rendering of the {}. a photo of the large {}. a rendering of a {}. a sketch of a {}. a photo of the {}. a cropped photo of the {}. a rendition of a {}. graffiti of a {}. a rendition of the {}. a photo of the small {}. a photo of one {}. a photo of the dirty {}. a photo of a dirty {}. a photo of a {}. a photo of many {}. a cropped photo of a {}. a photo of a large {}. a black and white photo of a {}. a painting of the {}. a photo of the nice {}. a photo of a small {}. a photo of the weird {}. a painting of a {}. a black and white photo of the {}. a low resolution photo of a {}. a dark photo of the {}. a dark photo of a {}. a doodle of the {}. a photo of the cool {}. a doodle of a {}. a low resolution photo of the {}. a photo of the hard to see {}. a blurry photo of the {}. a photo of a weird {}. a blurry photo of a {}`

Figure C.4: Example ImageNet images and the prompts by the uncertainty-based automatic prompt-filtering.

# D LIMITATIONS AND DISCUSSIONS

**Normal distribution with diagonal covariance would be insufficient?** One can argue that the uncertainty modeling power of PCME++ can be improved by relaxing the diagonal covariance condition. However, Oh et al. (2019) showed that if the dimensionality of the embedding space and the number of "hidden features" are the same (*e.g.*, if an image is the combination of two digits, then the number of potential latent features for each input is two), then the diagonal covariance condition can sufficiently capture the inherent uncertainty of the dataset. In practice, we use a very high dimensional embedding space (*e.g.*, 1024) that can sufficiently capture complex relationships between features. Also, in practice, if we relax the condition, the dimensionality of the $\log \sigma^2$ head output should be about 1M (= $1024 \times 1024$), which will require expensive computational budgets and large memory.

**Additional sampling is still required if we use other density functions.** The proposed probabilistic distance is defined in distribution-free: $\mathbb{E}_{\mathbf{Z}_v, \mathbf{Z}_t} \|\mathbf{Z}_v - \mathbf{Z}_t\|_2^2$. However, the closed-form solution (CSD) is specifically designed for normally distributed embeddings. If one needs probabilistic embeddings with different distributions, such as von Mises–Fisher distribution (Kirchhof et al., 2023) or Laplacian distribution (Warburg et al., 2023), CSD is no longer applicable. Instead, we can adapt any distribution to PCME++ by using a Monte Carlo approximation, *i.e.*, by computing $\frac{1}{n \times m} \sum_{z_v^i = z_v^1}^{z_v^n} \sum_{z_t^j = z_t^0}^{z_v^m} \|z_v^i - z_t^j\|_2^2$, where $z_v^i \sim \mathbf{Z}_v$ and $z_t^j \sim \mathbf{Z}_t$. This change will share the expensive computation issue of previous approaches (Oh et al., 2019; Chun et al., 2021), but the additionally introduced techniques in PCME++ for mitigating the loss saturation issue (*i.e.*, pseudo-positives and MSDA) will still be effective. Applying other probabilistic densities to PCME++ and discovering the effect of different distribution choices will be interesting future work.

**How does uncertainty help learning image-text representations?** As shown in the main experiments, the probabilistic approach is helpful for improving the retrieval performances, but the gaps are not significant (*e.g.*, Table 1 shows that in ViT-B/32, the gap between VSE$\infty$ and PCME++ with SWA is not significant). However, as shown in larger backbone experiments (ViT-B/16 and ViT-L/14) and noisy correspondence experiments (Table 2), PCME++ shows more generalizable performances compared to the existing state-of-the-art ITM methods with the same backbone. Furthermore, as shown in Section 3.4 and Section 3.5, the learned uncertainty by PCME++ shows high interpretability of the datasets as well as the controllability by the users when the rejection of the retrieved items is required. Thus, I believe that the uncertainty-aware learning paradigm and the learned uncertainty will be helpful for image-text matching problems and downstream tasks, such as zero-shot classification.

