# OpenReview forum: "Improved Probabilistic Image-Text Representations"
_ICLR.cc/2024/Conference — ICLR 2024 poster_

### Official Review · Reviewer_36mP · 2023-10-29

**Soundness:** 3 good
**Presentation:** 3 good
**Contribution:** 3 good
**Rating:** 6
**Confidence:** 5

**Summary:**

This paper presents an improved probabilistic representation to address issues including the presence of false negatives and sparse annotations in image-text matching.
Specifically, authors introduce two optimization techniques to enhance many-to-many matching, including pseudo-positives to prevent the loss saturation and mixed sample data augmentation for probabilistic matching

**Strengths:**

- well organized and easy to follow.
- the paper focuses on two important issues in image text matching: many-to-many matching and sparse annotations.

**Weaknesses:**

- Presented as Table 1 and 2, when ViT-B/32 was chosen as the backbone, the performance of PCME++ is inferior to VSE\infnite.
Additionally, the performances of P2RM and DAA published in the original paper actually won PCME with a large margin, while their performances presented here are lower than PCME.
The authors should make necessary explanations on these issues.
Most importantly, could the proposed PP and MSDA be applied to P2RM and DAA to enhance the many-to-many performance? If not, please make the necessary explanations; if yes, pls make comparisons.

- The datasets used in this submission including CxC and ECCV both come from MSCOCO, and thus the proposed method needs further verification on larger datasets like vision-language pretraining methods (CC3M dataset).
The authors are also recommended to make comparisons against several recent works focusing on uncertainty modeling [A] and noisy correspondence[B].


[A] MAP: Multimodal Uncertainty-Aware Vision-Language Pre-training Model, CVPR 2023.
[B] Deep Evidential Learning with Noisy Correspondence for Cross-modal Retrieval, ACM MM 2022.

**Questions:**

presented in weaknesses

---

> ### Author Response · Authors · 2023-11-16
>
> We thank the reviewer for their positive feedback and constructive comments. We will address all the raised concerns by the reviewer, and will revise our paper before next Tuesday.
>
> **[W1] PCME++ performance**
>
> Thanks for the question. First, we would like to emphasize that PCME++ is not inferior to VSE infty in all ranking-based metrics, such as mAP@R, R-Precision, and RSUM. It is because VSE infty is based on triplet loss using the hardest negative mining that is overfitted to recall@1 score as supported by Chun et al. 2022. Moreover, when the backbone size becomes larger, VSE infty shows inferior performances than PCME++ with a large margin in all metrics.
>
> We presume that it is because when the backbone size is sufficiently complex to be overfitted to the false negatives, deterministic methods easily fall to the overfitted solutions. We remark that the training dataset suffers from severe false negatives (88.2% of caption-to-image positives and 72.1% of image-to-caption positives are labeled as “negative” [Chun et al 2022]). On the other hand, the probabilistic methods are designed to handle many-to-many relationships, by less penalizing plausible matches (e.g., if an image-text pair is annotated as negative, but if the model predicts they are positive with high probability, the loss value becomes smoothed by its design – As discussed in Sec 2.3). It means that the effect of the FNs is saturated during training PCME++; hence it is profitable to be trained with large backbones.
>
> Second, both P2RM and DAA are basically an additional regularization to the existing loss function (P2RM employs less panelized additional triplet loss, and DAA employs CIDER-based extra information), usually triplet loss with the hardest negative mining. Furthermore, when we trained the models, we found that their methods are very sensitive to the hyperparameter selection, as shown in below:
>
> | .                  | mAP  | RSUM  |
> |--------------------|------|-------|
> | P2RM (a=1)         | 8.3  | 158.8 |
> | P2RM (a=0.2)       | 39.0 | 530.2 |
> | DAA (b=1)    | 39.2 | 530.9 |
> | DAA (b=0)    | 40.0 | 536.5 |
> | PCME++               | 40.1 | 537.0 |
>
> Perhaps, if we more carefully tune the hyperparameters for the methods, we would get slightly better performances than the reported numbers. However, when we tried to apply DAA and P2RM, we frequently faced training instability by choosing larger hyperparameters than our choice. Note that we use the official implementation of DAA for the comparison, and P2RM is our own implementation because there is no official code.
>
> Finally, the backbones used for our experiments differ from those of DAA, P2RM, and even PCME and VSE infty. Our implementation is a new and very strong setting based on CLIP backbones, showing the state-of-the-art retrieval performances against the existing visual semantic embeddings (VSEs) as shown in Tab C.1.
>
> **[W1] Applying PP and MSDA to comparison methods**
>
> We would like to emphasize that applying PP and MSDA to the triplet-based method is non-trivial. This is because the triplet loss is not designed for taking smooth labels. When we construct a triplet, we select an anchor sample (e.g., a visual embedding), its positive sample (e.g., its corresponding textual embedding), and its negative sample (e.g., one of the other textual embeddings). Here, for selecting the negative sample, triplet methods use the hardest negative mining, which uses the closest textual embedding as the negative. We argue that as there are abundant false negatives, it can harm the performances. The hardest negative mining is, therefore, very sensitive to the choice of batch size, as shown by VSE++ (https://github.com/fartashf/vsepp/issues/27).
>
> If we use a smooth label for a given pair, it is impossible to construct a triplet of <anchor, positive and negative>. For example, assume we set a match annotation of <image_a, caption_b> to 0.6 from 0.0. How can we build triplets using this annotation? Moreover, if we introduce mixed samples and mixed labels, the problem becomes more complex. How can we handle <image_a_mixed_b, caption_a> and <image_b_mixed_a, caption_b> using a triplet relationship? Therefore, it is impossible to apply PP and MSDA for triplet-based methods.
>
> One of our technical contributions is allowing smooth labels and mixed samples by designing a pairwise loss that is not affected by the other data samples. As shown in Fig 2., each loss computation of PCME++ is independent on the other pairs, while triplet loss or batch-wise contrastive loss is dependent on the relationships of other pairs.

---

> > ### Author Response · Authors · 2023-11-16
> >
> > **[W2] Larger datasets**
> >
> > We already provided the results of our model trained on the RedCaps dataset that contains 12M+ image-text pairs (Table 5). It is four times larger than the CC3M dataset. We compare the RedCaps pre-trained PCME++ and InfoNCE (i.e., CLIP) on zero-shot ImageNet classification. As shown in Table 5, we show that the probabilistic representation has a huge benefit when we fully utilize the knowledge from the learned uncertainty. Note that our main goal is not to propose a new vision-language pre-train model, but to examine the possibility of the uncertainty-aware methods on large-scale pre-training.
> >
> > **[W2] Other comparison methods**
> >
> > Thanks for the references. First, we would like to mention that MAP is an out-of-scope work of ours. It is because we only compare with the visual semantic embedding (VSE) methods that have separated visual and textual encoders, directly trained on the target dataset (i.e., MS-COCO). On the other hand, MAP is based on a cross-attention Transformer, which is not suitable for retrieval tasks. It should compute all pairwise similarity in the dataset, namely 5000 * 25000 = 125M forward time for measuring COCO-5K retrieval performances (it usually takes more than tens of hours with a single GPU), while VSE methods only need to forward 5000 + 25000 = 30k forward paths. However, we agree that MAP is a good reference that shows the effectiveness of uncertainty-aware learning in vision-language models. We will add a related discussion.
> >
> > We also thank to the reviewer for introducing DECL [B]. DECL-SGRAF shows 69.6 COCO 1K R@1, 49.4 COCO 5K R@1, and 518.2 RSUM in 20% noise, while PCME++ shows 71.6, 50.4, and 525.6, respectively. In 50% noise, DECL-SGRAF shows 68.3 COCO 1K R@1, 46.8 COCO 5K R@1 and 513.5 RSUM, while PCME++ shows 67.6, 45.5, 511.0, respectively. It is encouraged that although our method is not specifically designed for noisy correspondence, PCME++ shows better (in 20%) or comparable (in 50%) performances against the noisy correspondence-targeted methods without bells and whistles. We will add the numbers in Table 2 and related discussions.

---

> > > ### Author Response · Authors · 2023-11-21
> > > **Revision has been uploaded**
> > >
> > > Dear Reviewer 36mP,
> > >
> > > Thanks for your constructive and valuable comments on our paper. We have revised our paper to address the reviewer's concerns.
> > >
> > > - **[W1] PCME++ performance**: We clarify the details in Section 3.1, including we tried to search the hyperparemeters of DAA and P2RM, why the performance gap becomes larger when the backbone is scaled up
> > > - **[W1] Applying PP and MSDA to comparison methods**: We clarify the details in Appendix A.5., and cite the section in Section 2.4.
> > > - **[W2] Larger datasets**: We clarify the size of RedCaps in Section 3.5.
> > > - **[W2] Other comparison methods**: We added DECL in Table 2. As DECL does not provide the official checkpoints for 20% and 50% noises, we only report the COCO R@1 scores. We also add a discussion related to MAP in Appendix C.1.
> > >
> > > Please let us know if the reviewer thinks the revised version is insufficient. We will revise the paper as soon as possible.

---

> ### Author Response · Authors · 2023-11-22
> **Any follow-up question?**
>
> Dear Reviewer 36mP,
>
> We sincerely appreciate your efforts and time for the community. As we approach the close of the author-reviewer discussion period in 20 hours, we wonder whether the reviewer is satisfied with our response. It will be pleasurable if the reviewer can give us the reviewer's thoughts on the current revision to give us an extra valuable chance to improve our paper. We summarized our revision in the "Revision summary" comment and "Revision has been uploaded", where the latter is specially personalized to the reviewer's concerns.
>
> Again, we thank the reviewer's valuable commitment and their help to strengthen our submission. We will address all the raised concerns by reviewers if there remain any.

---

> > ### Comment · Reviewer_36mP · 2023-11-23
> > **Sincerely thanks for your detailed response.**
> >
> > Thanks for your detailed response.
> > My concerns are addressed and I have raised my score.
> >
> > Best

---

> > > ### Author Response · Authors · 2023-11-23
> > >
> > > Dear Reviewer 36mP,
> > >
> > > We deeply thank you for your time and dedication. We are pleased to hear that the reviewer's concerns are addressed.

---

### Official Review · Reviewer_RNqC · 2023-10-30

**Soundness:** 3 good
**Presentation:** 4 excellent
**Contribution:** 3 good
**Rating:** 8
**Confidence:** 3

**Summary:**

- This paper discusses some inherent issues with existing Image-Text Matching (ITM) methods and datasets, and provides analysis from the probabilistic perspective,

 - To solve the aforementioned issues, the authors propose Improved Probabilistic Image-Text Representations (PCME++) by introducing a closed-form probability distance to reduce the sampling costs of PCME and adopting two training techniques including pseudo-positive matches and mixed sample data augmentation.

 - Extensive experiments across abundant VL datasets show the effectiveness of PCME++. Moreover, primitive results on extending PCME++ to other scenarios are shown.

**Strengths:**

- Serving as its major motivation, this paper analyzes the many-to-many nature and other inherent issues in the Image-Text Matching task. Focusing on this fundamental Vision Language (VL) downstream task, this work addresses shortcomings of previous uncertainty-based methods such as PCME.

 - This presentation of this work is clear and easy to follow. In particular, the figures, such as Figure 2, are informative in illustrating important contexts.

 - The authors provide solid experimental results to support the effectiveness of PCME++. In addition to the evaluations on MS-COCO, CxC, and ECCV Caption with sota ITM methods, extensions to noisy correspondence and uncertainty-based prompt-tuning are conducted, which further demonstrate the advantage of PCME++.

**Weaknesses:**

- About the advantage of PCME++, the authors mentioned the advantage of PCME++ when scaling-up backbones, which might be lacking further discussions. (See Questions)

**Questions:**

- In Table 1, experimental results on different backbones are provided. While methods like VSE and InfoNCE show performance degradation, PCME and PCME++ show consistent improvements. Does this phenomenon relate to specific advantages of probabilistic modeling? (and How)

---

> ### Author Response · Authors · 2023-11-16
>
> We thank the reviewer for their positive evaluation and valuable comments. We will address all the raised concerns by the reviewer, and will revise our paper before next Tuesday.
>
> **[W1, Q1] Lack of discussions of the backbone scale-up**
>
> Thanks for your comment. The discussion related to the question is in Sec 3.2. Main results. “We conjecture that it is because the non-probabilistic counterparts are easily overfitted toward FNs when the model has more complexity, particularly with the hardest negative mining strategy.” Here, we provide more details beyond this explanation.
>
> As shown in Fig 1, the training dataset already suffers from severe false negatives (88.2% of caption-to-image positives and 72.1% of image-to-caption positives are labeled as “negative” [Chun et al 2022]). As the backbone becomes more complex and larger, we presume that the backbone complexity is sufficiently large to capture the noisy FNs in the dataset. Moreover, VSE infty uses a triplet loss with the hardest negative mining, which will make the effect of FNs more significant. On the other hand, the probabilistic methods are designed to handle many-to-many relationships by less penalizing plausible matches (e.g., if an image-text pair is annotated as negative, but if the model predicts they are positive with high probability, the loss value becomes smoothed by its design – As discussed in Sec 2.3). It means that the effect of the FNs is saturated during training PCME and PCME++. However, as we discussed in Sec 2.3., at the same time, it can cause a fast loss saturation for PCME. We address the problem by proposing two techniques: pseudo-positives and mixed data sample augmentation. Note that it is non-trivial to employ the techniques to triplet-based metric learning methods (VSE infty) and batch-wise contrastive learning methods (InfoNCE).

---

> > ### Author Response · Authors · 2023-11-21
> > **Revision has been uploaded**
> >
> > Dear Reviewer RNqC,
> >
> > Thanks again for your positive feedback. We are highly encouraged by your comments. We have uploaded the revised paper, including more discussions of the backbone scale-up. We highlight the revised contents in magenta.
> >
> > - **[W1, Q1] Lack of discussions of the backbone scale-up**: We add more detailed discussion in Section 3.2.
> >
> > Please let us know if the reviewer thinks the revised version is insufficient. We will revise the paper as soon as possible.

---

> ### Author Response · Authors · 2023-11-22
> **Any follow-up questions?**
>
> Dear Reviewer RNqC,
>
> We deeply thank you for your dedication to the community. As we mentioned before, the positive feedback from the reviewer highly encouraged us. We again sincerely appreciate your valuable comments. As the author-reviewer discussion period is closed in 20 hours, we wonder whether the revised paper is still satisfactory to the reviewer. It will be pleasurable if the reviewer can give us the reviewer's thoughts on the current revision to give us an extra valuable chance to improve our paper. We summarized our revision in the "Revision summary" comment and "Revision has been uploaded", where the latter is specially personalized to the reviewer's concerns.
>
> Again, we thank the reviewer's valuable commitment and their help to strengthen our submission. We will address all the raised concerns by reviewers if there remain any.

---

### Official Review · Reviewer_XAP9 · 2023-10-31

**Soundness:** 3 good
**Presentation:** 3 good
**Contribution:** 2 fair
**Rating:** 6
**Confidence:** 3

**Summary:**

This paper presents an improved Probabilistic Cross-Modal Embeddings (named PCME++) by introducing a new probabilistic distance, incorporates of pseudo-positives to prevent the loss saturation problem, and introduce the data augmentation technique. Experimental results demonstrate the effectiveness of the proposed model.

**Strengths:**

1. The representation is good and easy to follow.

2. The robustness of PCME++ is evaluated under noisy image-text correspondences.

3. The most of the figures in this paper is informative, especially Figure1 and Figure5.

**Weaknesses:**

1. This work is deeply coupled with PCME( Chun et al. (2021)), which might limiting the inspiration and extensibility for other work.

2. As shown in Figure 5, the visualization could show the uncertainty of learned embeddings of visual features and textual features. However, the proposed closed-form sampled distance (CSD) could only simply measure the uncertainty but not the area of the overlap between two modality. I hope the authors could make more analysis.

3. Some techniques such as Mixup (Zhang et al., 2018) or CutMix (Yun et al., 2019) are proposed in the previous works, and the loss functions are shared with PCME( Chun et al. (2021)). Please clarify your contribution and improvements.

If my concerns are solved, I would like to raise my score.

**Questions:**

1. Figure 2 is not easy-understood. Please make interpretations about it and formulate the comparison losses.

2. The motivation of this paper is "PCME suffers from expensive computations due to Monte Carlo approximation and fast loss saturation". However, the comparison or analysis about computation costs is not presented in the experiments.

---

> ### Author Response · Authors · 2023-11-16
>
> We thank the reviewer for their positive feedback and constructive comments. We will address all the raised concerns by the reviewer and will revise our paper before next Tuesday.
>
> **[W1] Inspiration and extensibility for other work**
>
> We would like to emphasize that our work is an independent image-text matching work, not depending on specific methods. PCME++ is not strongly coupled with PCME. We will discuss it in more detail in the next bullet (**Similarity between PCME loss function and PCME++ loss function**). We presume that the reason why the reviewer feels that PCME++ limits the inspiration and extensibility for other work is that other approaches are usually based on deterministic methods. The technical details for learning a probabilistic method are somewhat difficult to be applied to deterministic methods. However, our work brings attention to the importance of the false negatives in the Image-Text matching training dataset (as pointed out by Reviewer RNqC). We explain that the FNs can be problematic for deterministic methods, especially for larger backbones, such as ViT-L/14. We do not believe that a probabilistic approach is the only method to tackle FNs. We believe that our work can give inspiration to other ITM methods to tackle the FN problem raised in our work.
>
> **[W1, W3] Similarity between PCME loss function and PCME++ loss function**
>
> We would like to emphasize that the similarity between PCME and PCME++ is due to the nature of the probabilistic embeddings (probemb). The main goal of probemb is to map an input to a random variable (r.v.) rather than a deterministic vector. We would like to emphasize that the core ideas of other probembs are also similar to each other, for example, Oh et al. 2019, Sun et al., 2020, Shi and Jain, 2019, Chang et al., 2020, SilnovaPark et al., 2022a et al., 2020 and Neculai et al., 2022. PCME (Chun et al. 2021) and our method, therefore share the property of probemb: we train an encoder that maps an input to a mean vector and a variance vector and train the encoder by maximizing the negative loglikelihood (NLL) using the extracted distributions.
>
> PCME optimizes NLL by defining “matching probability”:
> $p(m | x_v, x_t) = \mathbb E_{Z_v, Z_t} \text{sigmoid} (-a \|\| Z_v - Z_t \|\| + b) \approx \frac{1}{J^2} \sum_j^J \sum_{j^\prime}^J \text{sigmoid} ( - a \|\| z_v - z_t \|\| + b)$
>
> On the other hand, PCME++ optimizes the NLL using binary cross-entropy loss:
> $p(m | x_v, x_t) = \text{sigmoid} (-a \mathbb E_{Z_v, Z_t} \|\|Z_v - Z_t\|\|^2 + b)$
>
> Although they look somewhat similar, PCME takes the expectation over the sigmoid of the pairwise distance of r.v.s, while PCME++ takes the expectation over the pairwise distance of r.v.s before taking the sigmoid. The latter has two benefits over the former: (1) our form has a closed-form solution, as shown in Eq (1), while PCME cannot. (2) our form can be naturally adopted into the binary cross entropy loss function, which is known to be stable and perform well in large-scale training [1].
>
> Therefore, we argue that PCME and PCME++ use different loss functions in terms of the NLL formulation and the probabilistic distance (PCME uses the Monte-Carlo approximation of “matching probability”, and PCME++ uses the closed-form sampled distance, CSD).
>
> [1] ResNet strikes back: An improved training procedure in timm
>
> **[W3] Augmentation methods are already proposed in previous works**
>
> Although we do not propose a new MSDA, our contribution lies in applying MSDA to the relational datasets. For example, applying MSDA to classification is straightforward because the mixed sample does not affect the other samples in the mini-batch. However, in the relational training objectives, such as triplet loss or contrastive loss, a mixed sample affects the other samples in the batch as well. Especially, the triplet loss is impossible to handle MSDA, because the core concept of MSDA is the smooth label but the triplet loss cannot handle smooth label, because it has to construct a triplet of the selected sample, the positive sample, and the negative sample. It is non-trivial to define positive and negative samples when the label is smoothed (Fig 2a). Similarly, a batch-wise contrastive loss, such as InfoNCE, is also a little bit tricky to control the effect of smooth labels (Fig 2b) because the mixed samples are combined in the denominator term of the InfoNCE softmax. On the other hand, our pairwise contrastive loss can directly apply smooth labels because each loss computation is invariant to the other samples, but only the given pairs affect the loss computation (Fig 2d).
>
> Overall, we do not argue that we newly propose MSDA (CutMix and Mixup), but we argue that we first apply MSDA to image-text matching methods that are based on relational datasets, not instance-level datasets (e.g., image-only datasets).

---

> > ### Author Response · Authors · 2023-11-16
> >
> > **[W2] Area of overlap between two modalities vs. CSD**
> >
> > Thanks for the valuable feedback. First, we would like to emphasize that we never mentioned the area of overlap between the two modalities in the paper. The purpose of Fig 5 is to show how the learned embedding space can successfully capture the inherent ambiguity in the dataset. Qualitatively, PCME++ embedding space captures the ambiguity of the many-to-many relationships despite the false negatives.
> >
> > Second, the overlap between the two distributions itself is not directly related to the uncertainty. Conceptually, the overlap between two Gaussian distributions can be represented as the Bhattacharyya coefficient (or Bhattacharyya distance). Here, we recall the CSD’s main property (b) discussed in Sec 2.2: if the match $m_{vt}$ is certain, then $Z_v$ and $Z_t$ have small variances. As we discussed in Sec 2.2, a similarity function that measures whether two distributions are the same or not cannot satisfy the property because it cannot handle the case when the distributions have the same $\mu$, but larger $\sigma$. There is no motivation to reduce the size of $\sigma$ using the distance. As shown in Table 4, Bhattacharyya distance is not effective probabilistic distance for learning probemb as much as CSD. On the other hand, the learned embedding space by CSD-trained PCME++ is a reasonable probabilistic space. Table C.5 shows that even though we use Wasserstein distance as the similarity function of retrieval, the overall precision-based retrieval performances are almost preserved; it means that the probabilistic space learned by PCME++ is a sound metric space of Wasserstein distance.
> >
> > Third, instead of focusing on the overlap between two distributions of the learned embedding space, we would like to suggest how CSD can learn the embedding space shown in Fig 5. We recall the properties of our desired probabilistic embedding space: (1) if the correspondence between a given image and caption is certain (i.e., they are certainly positive or negative), then the variance of each instance should be small, (2) if the correspondence is uncertain (i.e., the match is sometimes positive and sometimes negative. It can be happened due to the false negatives in the dataset as shown in Fig 1 and Sec 2.1), then the variance of each instance should be large. As we mentioned before, CSD can give proper supervision for the desired embedding space. For example, let’s approximate the plane figures in Fig 5 to a visual embedding $v$ and the plane captions to a textual embedding $t$ (because they have very high semantic similarity). In this case, by choosing different image-caption pairs, the supervision between $v$ and $t$ can be either positive or negative because our training dataset only has one-to-one positive correspondences. In this case, our objective function enforces the CSD between the matches to be larger until the penalty for the positive supervision and the negative supervision are balanced. As shown in Fig 5, the visual embeddings and textual embeddings are aligned into almost the same place by making the $\mu$ distance closer but penalizing the uncertain supervision by enlarging $\sigma$.
> >
> > **[Q2] Computation cost comparison**
> >
> > Thanks for your comment. The computational cost of PCME depends on the number of MC samples $J$, because it needs to compute $O(J^2)$ pairwise distances between all samples. When we use the same setting of the paper ($J=8$), we observe that PCME++ 25 epoch training takes 106,311 secs (1 day and 5 hours), while PCME 25 epoch training takes 141,694 secs (1 day and 15 hours) on a single V100 GPU. Overall, PCME needs 33% more training time compared to PCME++. Note that if we increase the sampling size $J$, the gap becomes larger. Another issue of the PCME sampling is that we need more memory size when computing the Monte Carlo approximation for a larger sampling size. Overall, PCME needs more forward time than PCME++ (33% more), and more memory size than PCME++ (on average, 18% more, but it is not a rigorous comparison because PCME has higher peak memory usage).
> >
> > **[Q1] Clarity**
> >
> > We will revise Figure 2 following the reviewer’s comment. The main motivation of Figure 2 is to compare how different objective functions handle mini-batch samples. As we mentioned in **[W3] Augmentation methods are already proposed in previous works**, the figure shows that it is somewhat non-trivial to apply MSDA to the existing methods because a mixed sample affects across the entire relationships between images and captions.

---

> > > ### Comment · Reviewer_XAP9 · 2023-11-20
> > > **Thanks for the author's response.**
> > >
> > > Thanks for your reply. Some of my concerns have been solved. The discussion in the response should be added into the main paper (if accepted), such as the loss function, computation cost comparison, and the novelty. I'd like to raise my score to weak accept.

---

> > > > ### Author Response · Authors · 2023-11-21
> > > > **Revision has been uploaded**
> > > >
> > > > Dear Reviewer XAP9,
> > > >
> > > > We truly appreciate your dedication and thoughtful review. We have uploaded the revised paper. We highlight the revised contents in magenta. Due to the page limitation, we move Figure 2 in the Appendix.
> > > >
> > > > - **[W1] Inspiration and extensibility for other work**: We re-emphasize our contribution in Conclusion section.
> > > > - **[W1, W3] Similarity between PCME loss function and PCME++ loss function**: We clarify the details in Appendix A.3. We cite the Appendix section in Section 2.3
> > > > - **[W3] Augmentation methods are already proposed in previous works**: We clarify the details in Appendix A.5., and cite the section in Section 2.4.
> > > > - **[W2] Area of overlap between two modalities vs. CSD**: The Appendix C.4. covers the discussion. It is cited in Section 3.4.
> > > > - **[Q2] Computation cost comparison**: In Section 2.2, we specify PCME++ is 33% faster than PCME. More detailed comparisons are in Section A.3.
> > > > - **[Q1] Clarity**: We tried to update the figure itself, but we found that directly writing the objective functions is not proper for the figure, because we intended to cover general objective functions in Figure 2 (now in Figure A.3.). Instead, we modify the caption of the figure to be more detailed and plenty.
> > > >
> > > > Please let us know if the reviewer thinks the revised version is insufficient. We will revise the paper as soon as possible.

---

### Author Response · Authors · 2023-11-21
**Revision summary**

We truly appreciate all the reviewers and the chairs for their commitment to the community and thoughtful reviews. We are highly encouraged that the reviewers found that our analyses on the many-to-many nature of ITM methods are a strength (Reviewer RNqC, 36mP), the experimental results are solid (Reviewer XAP9, RNqC) the figures are informative (Reviewer XAP9, RNqC), our paper is well organized and easy to follow (Reviewer XAP9, RNqC, 36mP). We luckily have a chance to make our work stronger, with constructive and valuable comments from the reviewers. We have addressed all the raised concerns by the reviewers in the revised paper. The revised contents are highlighted in magenta.

Here, we summarize the major changes in detail:

- In Section 3.2, we clarify the rationale of why PCME++ is successfully scaled up with a larger backbone, while the deterministic counterparts cannot (Reviewer RNqC, 36mP)
- We clarify why MSDA is not applicable to previous methods in Appendix A.5 and cited it in Section 2.4 (to address the concerns raised by Reviewer XAP9, 36mP)
- We clarify the computational advantage of PCME++ over PCME in Section 2.2 and A.3 (RNqC)
- We clarify the difference between PCME++ and PCME in terms of the objective functions in Section A.3 (RNqC)
- We clarify the meaning of Figure 4 (previously Figure 4) in Section C.4. It is cited in Section 3.4.
- In Section 3.1, we clarify that we tried our best to implement the comparison methods in Section 3.1, for example, we tuned the optimization hyperparameters based on VSE infty ViT-B/32, and we searched the hyperparameters for DAA and P2RM, following our validation strategy. (Reviewer 36mP)
- We add more comparisons with DECL in Table 2. We also add a discussion related to MAP in Section C.1 (Reviewer 36mP)
- We clarify the size of RedCaps in Section 3.5 (Reviewer 36mP)
- We re-emphasize our contribution: bringing attention to the importance of the FNs in ITM training datasets (Reviewer XAP9)
- We make the caption of Figure A.3. (previously Figure 2) for better clarity.

Please feel free to ask anything if the reviewers think the revised paper is insufficient. We are open to discussion and will address all the concerns of the reviewers.

---

### Meta-Review · Area_Chair_yYFP · 2023-12-07

**Metareview:**

The paper discusses the inherent issues in ITM, and proposes a novel probabilistic method PCME++ incorporating a closed-form probability distance and pseudo-positive matches. The authors performed an extensive set of experiments on multiple VL datasets showing the effectiveness of PCME++. The work explores the inherent but underexplored issues of ITM and this attempt seems timely. Since such issues are also present in other tasks, similar ideas can be adopted for different tasks.

While the reviewers initially raised concerns about limited comparisons and explanations with existing methods, the reviewers acknowledged that the concerns are addressed by the author rebuttal. As a result, all three reviewers reached a positive consensus leading the AC to recommend the acceptance of the paper.

**Justification For Why Not Higher Score:**

Despite the strong motivation and the novelty of the method, the improvement is marginal compared to the existing work limiting its potential impact.

**Justification For Why Not Lower Score:**

All the reviewers recommend the acceptance acknowledging the contributions of the paper.

---

### Decision · Program_Chairs · 2024-01-16

Accept (poster)